

# Symmetries and topological operators, on average

**Andrea Antinucci, Giovanni Galati, Giovanni Rizi and Marco Serone**

SISSA, Via Bonomea 265, I-34136 Trieste, Italy
INFN, Sezione di Trieste, Via Valerio 2, I-34127 Trieste, Italy

## Abstract

We study Ward identities and selection rules for local correlators in disordered theories where a 0-form global symmetry of a QFT is explicitly broken by a random coupling $h$ but it re-emerges after quenched average. We consider $h$ space-dependent or constant. In both cases we construct the symmetry operator implementing the group action, topological after average. In the first case, relevant in statistical systems with random impurities, such symmetries can be coupled to external backgrounds and can be gauged, like ordinary symmetries in QFTs. We also determine exotic selection rules arising when symmetries emerge after average in the IR, explaining the origin of LogCFTs from symmetry considerations. In the second case, relevant in AdS/CFT to describe the dual boundary theory of certain bulk gravitational theories, the charge operator is not purely codimension-1, it can be defined only on homologically trivial cycles and on connected spaces. Selection rules for average correlators exist, yet such symmetries cannot be coupled to background gauge fields in ordinary ways and cannot be gauged. When the space is disconnected, in each connected component charge violation occurs, as expected from Euclidean wormholes in the bulk theory. Our findings show the obstruction to interpret symmetries emergent after average as gauged in the bulk.


doi:10.21468/SciPostPhys.15.3.125

# 1  Introduction

Global symmetries constitute an indispensable tool for studying physical systems, especially when the dynamics cannot be analyzed using exact techniques. The idea of symmetry is sometimes vaguely stated and often confused with slightly different concepts, such as *selection rules*. While these two ideas are often connected, they are logically distinct. Moreover, symmetry must be distinguished from gauge invariance, which is a redundancy of a certain formalism rather than an actual symmetry of the physical system.

For these reasons, the idea of symmetry has recently been made more precise and intrinsic through the notion of *topological operators* (see [1] for a modern treatment). These are extended operators or defects in Quantum Field Theory (QFT) supported on submanifolds in space-time. Their dependence is purely topological: small deformations of the support do not change the correlation functions, but when they pass a charged operator, it undergoes a symmetry transformation. The topological nature implies that, in any quantization scheme, if they are placed on a space-like slice, they become operators on the Hilbert space commuting with the Hamiltonian, recovering the standard notion of symmetry in quantum systems [2]. In Euclidean QFTs, which we will focus on, we do not distinguish between operators and defects, so we will refer to such operators generically as topological operators.

In finite volume, when a QFT has invertible topological operators implementing a group of symmetries, they imply selection rules on correlation functions, and the same remains true in infinite volume if the symmetry is not spontaneously broken. More importantly, the formalization of symmetry in terms of topological operators allows several interesting generalizations. These include higher-form symmetries (starting from [1,3]), higher groups (introduced in the physics literature in [4–6]), non-invertible symmetries in two-dimensions (see e.g. [7,8]) and more recently also in higher dimensions after [9,10], leading to the notion of higher-categorical symmetries [11,12]. These developments are based on the notion of topological operators introduced in [1] and allowed the extension of the concept of 't Hooft anomalies to discrete and generalized symmetries and new constraints on the dynamics of strongly coupled systems (see for instance [13–16]).

The aim of this work is to extend the formalization of symmetries of [1] to QFTs where the interactions are randomly distributed, for the case of 0-form global symmetries. We believe that a more systematic treatment of symmetries in QFTs of this kind can be useful, given the notorious difficulties in treating such systems. There are two relevant possibilities considered here.

1. The random couplings $h(x)$ vary in space and are distributed according to a probability functional $P[h]$.

2. The random couplings $h$ are constant and drawn from a probability density $P(h)$.

Scenario 1. is relevant for statistical mechanical systems with impurities or disorder (for a review see [17]). There are two main variants of disorder QFT: *quenched* if the impurities are treated as external random sources and *annealed* if the impurities are taken dynamical. Physically the two situations depend on the time scale we are looking at. At extremely long time-scales, where the entire system reaches equilibrium, we should take the impurities dynamical. Since impurities have very long thermalizations time scales, quenching is useful for time-scales where the system essentially thermalizes, with the impurities taken fixed. In the quenched case, the properties of the QFT will of course depend on the impurities. If we assume that impurities are random, possible observables are taken by averaging over the impurities with the chosen distribution. In a lattice formulation an impurity is modelled by an interaction which is different at any site, and its presence is unpredictable. In the continuum limit it is often the case that we can describe such systems as the average over an ensemble of field theories where the coupling constants are space dependent. Particularly interesting is the case of the Ising model perturbed with a random magnetic field (dubbed as random field Ising model) [18] or with a random interaction between nearby spins (dubbed as random bond Ising model) [19]. See e.g. [20–23] for recent works on these models.

Scenario 2. is relevant for quantum gravity and has received significant attention lately. The connection between averaging and euclidean gravity path integrals dates back to [24,25] in association to Euclidean wormholes. In the context of the AdS/CFT correspondence [26–28], the connection has been invoked in [29] as a possible way to interpret from a boundary point of view the origin of interactions between disconnected components of a boundary theory induced by bulk Euclidean wormholes (factorization puzzle). Further elaborations with concrete examples appear in [30]. Ensemble averaging features also in the Sachdev-Ye-Kitaev (SYK) model [31–33]. A concrete connection has recently been made in [34], where it has been shown that the sum over geometries in Jackiw-Teitelboim gravity [35,36] with $n$ disconnected boundaries is dual to the ensemble average of an $n$-point correlation function in a matrix model. Other notable examples of ensemble averaging after [34] include averages over free compact bosons in 2d [37,38] (see also e.g. [39–43] for related studies and generalizations), averages over OPE coefficients in effective 2d CFTs [44,45], averages over the gauge coupling in 4d $\mathcal{N} = 4$ super Yang-Mills theory [46].

In both scenarios 1. and 2. we focus on correlation functions of local operators with *quenched disorder* averaging. These include averages of products of correlators, which are effectively independent observables. In disconnected spaces, when $h$ is constant, also averaged single correlators can lead to averages of products of correlators, which is the mechanism leading to the factorization puzzle in the context of AdS/CFT. In order to distinguish scenario 1. from scenario 2. we dub the first as "quenched disorder" and the latter as "ensemble average", but it should be kept in mind that quenching is involved also in scenario 2.

We start from a *pure theory*, that is an ordinary QFT with no disorder, and deform it with a certain local interaction. In the quenched disorder case the strength of this interaction varies from point to point, while it is constant in ensemble average. In both cases the interaction

can break part of all of the global symmetries of the pure system, so that each specific realization generically has less symmetries and less predictive power than the pure theory. On the other hand it has been noticed in several examples that symmetries of the pure systems can be recovered after the average on the coupling is taken into account. These statements are mostly based on the observation that the averaged system satisfies selection rules which are not enjoyed by the generic specific realization. Intuitively speaking, even if the random coupling breaks the symmetry, this re-emerges provided we average over all the ensemble in a sufficiently symmetric way (in a sense to be clarified). For simplicity, in what follows we refer to such symmetries as *disordered symmetries* and *averaged symmetries* respectively in the context of quenched disorder and ensemble average. Note that this is distinct from the notion of emergent symmetries used in pure QFTs when a symmetry is approximately conserved in the IR. In the disorder case the symmetry is *exact* at all energy scales, but only on average.[1]

We will review these kind of arguments from a spurionic point of view at the beginning of section 2.2 for quenched disorder, and in section 5.1 for ensemble average, deriving under which condition the selection rules of the pure theory are satisfied after the average.

This is still an imprecise information since, as we emphasized, having a global symmetry is stronger than just observing the validity of some selection rule. This is crucial in order to get stronger dynamical constraints implied by 't Hooft anomalies, and eventually gauging the symmetry. Our goal is to clarify the sense in which these systems recover the symmetry, aiming to construct the analog of topological operators for both quenched disorder and ensemble average QFTs.

Sections 2, 3 and 4 focus on quenched disordered systems. We consider theories defined in the continuum and admitting a description in terms of an action (Hamiltonian) obtained from that of the pure theory $S_0$ by adding a local operator $\mathcal{O}_0(x)$ with a space-time dependent coupling $h(x)$:

$$S[h] = S_0 + \int d^d x\, h(x)\mathcal{O}_0(x).$$ (1)

This is what we will call a specific realization. Correlation functions of local operators $\mathcal{O}_i$ for a given value of $h(x)$ are computed by a path integral:

$$\left\langle \mathcal{O}_1(x_1)\cdots\mathcal{O}_k(x_k)\right\rangle = \frac{\int \mathcal{D}\mu\, e^{-S[h]}\,\mathcal{O}_1(x_1)\cdots\mathcal{O}_k(x_k)}{\int \mathcal{D}\mu\, e^{-S[h]}}.$$ (2)

Given a probability functional $P[h]$, a set of observables of the disordered system are the averaged correlation functions

$$\overline{\left\langle \mathcal{O}_1(x_1)\cdots\mathcal{O}_k(x_k)\right\rangle} = \int \mathcal{D}h\, P[h]\left\langle \mathcal{O}_1(x_1)\cdots\mathcal{O}_k(x_k)\right\rangle,$$ (3)

or more generally the averages of products of correlators

$$\overline{\prod_{j=1}^{N}\left\langle \mathcal{O}_1^{(j)}(x_1^{(j)})\cdots\mathcal{O}_{n_j}^{(j)}(x_{k_j}^{(j)})\right\rangle}.$$ (4)

The starting point for systematizing global symmetries of the disordered system which are not enjoyed by the specific realizations is to derive Ward identities for the averaged correlators. We do this in section 2.2, starting from the simplest case of continuous 0-form invertible symmetries. The Noether current $J^\mu$ associated to the symmetry of the pure theory is no longer

---

[1] We can also have emergent symmetries in both senses, namely emerging after average and in the IR. We will discuss this case in section 4.

conserved in the specific realizations if $\mathcal{O}_0(x)$ is charged under the symmetry. However we find that the *shifted current* in (37) leads to standard Ward identities (41) for averaged single correlators and the less standard identities (45) for averages of products of correlators.

In order to generalize our results to discrete symmetries, where a Noether current is unavailable, in section 2.3 we construct the symmetry operators, topological on average, which implements the finite group action. This is not as easy as in pure theories because of the disorder. The topological operator $\widetilde{U}_g$ is a complicated power series of integrated currents which however can be resummed to give the simple expression

$$\widetilde{U}_g = U_g \langle U_g \rangle^{-1}, \tag{5}$$

where $U_g$ is the topological operator of the pure theory. Its action on averages of simple correlators is given in (58). In products of correlators (4) the operator $\widetilde{U}_g$ is topological on average only if inserted in all the correlators involved, as in (61). This characterizes intrinsically the disordered symmetries and implies somewhat exotic selection rules which are weaker with respect to symmetries not broken by the random interactions.

The Ward identities satisfied by $\widetilde{U}_g$, when the latter is supported on a compact surface $\Sigma^{(d-1)}$ are valid regardless of how the symmetry is realized on the vacuum. When the symmetry operator is well defined also on infinite surfaces the disordered symmetry is not spontaneously broken and implies selection rules. The same is not true for spontaneously broken symmetries, we will briefly discuss this situation in the final section 6.

Beyond selection rules, our analysis allows us to show that disordered symmetries (both continuous and discrete) can be coupled to external backgrounds, can be gauged, and can have 't Hooft anomalies, precisely like ordinary symmetries. We also argue that a symmetry of a pure system with a 't Hooft anomaly, when it reappears as disordered symmetry, enjoys the same 't Hooft anomaly thus implying the same constraints on the dynamics, and that a possible higher-group structure of the underlying 0-form symmetry with higher-form symmetries of the pure theory is recovered after average due to the topological nature of the higher-group structure. Symmetry Protected Topological (SPT) phases [47], protected by what we denoted disordered symmetries, appeared already in condensed matter, see e.g. [48–55]. Our findings can possibly provide a different theoretical QFT-based framework for such phases of matter.

In section 3 the above results, derived directly from the disordered theory, are reproduced using the replica trick, the standard way to deal with theories of this kind. Disordered symmetries manifest themselves as standard symmetries in the replica theory, thus offering a conceptual different viewpoint on these kind of symmetries. Aside from providing a sanity check of the results, the replica theory allows us to also study another scenario: disordered symmetries emerging at long distances, discussed in section 4. The effect of the disorder can now lead to the more exotic selection rules (104) and (105). The phenomenon manifests in the replica theory as two irreducible representations of the replica symmetry transforming in different representations of the emergent disordered symmetry. As an interesting application of this result we consider the prime example of an emergent symmetry, conformal invariance, and we show that as a consequence of these exotic selection rules, a quenched disordered system can flow in the IR to a fixed point described by a Logarithmic conformal field theory (LogCFT) [56–60].

We analyze ensemble average in section 5. While the intuitive idea that the average restores the symmetry is still true, and selection rules apply (section 5.1), the status of the averaged symmetry is drastically different. A hint already comes from the replica trick: when applied with constant couplings, the replica theory is non-local, and even if the symmetry is manifest its Noether current is not a local operator. This is problematic for constructing a topological operator. Indeed, independently of the replica trick, we imitate the analysis done for disordered theories, and we get the exotic topological charge operator (134). This is not

really a co-dimension one operator, since it depends both on a closed surface $\Sigma^{(d-1)}$ and on a filling region $D^{(d)}$ such that $\partial D^{(d)} = \Sigma^{(d-1)}$. In particular the operator cannot be supported on homologically non-trivial cycles. Crucially, the operator $\widehat{Q}$ implies selection rules, because the second term in (134), when inserted on average of correlation functions of local operators, vanishes when integrated over the full space. If the space manifold is connected, there are only two possible filling regions of a homologically trivial $\Sigma^{(d-1)}$, and $\widehat{Q}$ is independent of the choice. On the other hand on a *disconnected* space there are several choices of filling region $D^{(d)}$, and the charge operator does depend on these choices. Nevertheless, we do have selection rules for averages of correlators, if one takes into account all the connected components of space, and we can construct operators (B.20) implementing the finite group action. In each connected component the selection rules can be violated, allowing the charges to escape from one component to the other. We have then the somewhat exotic situation of a 0-form symmetry in the sense of selection rules on correlation functions of local operators, but without having genuine topological operators (even after average). In contrast to ordinary symmetries and disordered symmetries in the quenched disordered case above, averaged symmetries cannot be coupled to background gauge fields in ordinary ways and hence cannot be gauged.

In section 5.3 we comment about the gravity interpretation of these results. Whenever the average theory admits a gravitational bulk dual, the local charge violation in presence of disconnected space has the natural interpretation in the bulk as charge violation induced by Euclidean wormholes configurations, as pointed out in [61–63]. The difficulty (impossibility) of gauging averaged boundary symmetries that we have found clarify why such symmetries cannot be identified with bulk gauge symmetries.

We conclude in section 6. In appendix A we work out some specific examples for concreteness, and in appendix B we explicitly construct the operator which implements the action of the group for averaged symmetries.

## 2 Symmetries in quenched disorder

In this section we study global 0-form symmetries in quenched disorder theories which arise only after the average. We start in section 2.1 by reviewing how Ward identities for ordinary 0-form symmetries are recast in terms of topological operators in pure QFTs. We generalize the analysis to theories with quenched disorder in section 2.2 and construct the topological operator implementing the symmetry group action in section 2.3. We then discuss 't Hooft anomalies and gaugings for both continuous and discrete disordered symmetries in sections 2.4 and 2.5.

### 2.1 Pure theories and explicit symmetry breaking

Consider a standard $d$-dimensional Euclidean QFT described by the action $S_0$. If this theory is invariant under some continuous symmetry group $G$, correlation functions of local operators must satisfy the usual constraints imposed by the Ward-Takahashi identities:

$$i\big\langle \partial_\mu J^\mu(x) \mathcal{O}_1(x_1) \ldots \mathcal{O}_k(x_k) \big\rangle = \sum_{l=1}^{k} \delta^{(d)}(x - x_l)\big\langle \mathcal{O}_1(x_1) \ldots \delta\mathcal{O}_l(x_l) \ldots \mathcal{O}_k(x_k)\big\rangle. \quad (6)$$

Here $J_\mu(x)$ is the Noether current[2] and $\delta\mathcal{O}_l(x_l)$ is the transformation of the local operator $\mathcal{O}_l$ under the action of the Lie algebra of $G$. For instance if $G = U(1)$ and $\mathcal{O}_l$ has charge $q_l$, then

---

[2]For convenience we define the Noether current as $\delta S = i \int \epsilon(x) \partial^\mu J_\mu$. Notice that this has an extra factor of $i$ with respect to the one obtained by Wick rotating the standard Minkowski current.

$\delta\mathcal{O}_l = iq_l\mathcal{O}_l$. Integrating over the full space $X^{(d)}$, the left hand side of (6) vanishes if $X^{(d)}$ has no boundary and the symmetry is not spontaneously broken, and we get selection rules on the correlators.

The modern approach [1] to interpret the same constraints consists in associating global symmetries to co-dimension one *topological operators* $U_g[\Sigma^{(d-1)}]$, $g \in G$, namely extended operators supported on some $(d-1)$-dimensional closed surface $\Sigma^{(d-1)}$, which are invariant under continuous deformations of their support. In the case of continuous symmetries such topological operators are simply[3]

$$U_g[\Sigma^{(d-1)}] = e^{i\alpha Q[\Sigma^{(d-1)}]}, \tag{7}$$

where $Q[\Sigma^{(d-1)}] = \int_{\Sigma^{(d-1)}} J_\mu n^\mu$ is the Noether operator which measures the charge enclosed within the region $D^{(d)}$ delimited by $\Sigma^{(d-1)}$ with $\partial D^{(d)} = \Sigma^{(d-1)}$. We can then write integrated Ward identities. For instance, if $G = U(1)$ we have

$$\left\langle Q[\Sigma^{(d-1)}]\mathcal{O}_1(x_1)\dots\mathcal{O}_k(x_k)\right\rangle = \chi(\Sigma^{(d-1)})\left\langle\mathcal{O}_1(x_1)\dots\mathcal{O}_k(x_k)\right\rangle, \tag{8}$$

with

$$\chi(\Sigma^{(d-1)}) = \sum_{l,x_l \in D^{(d)}} q_l. \tag{9}$$

The integrated Ward identity can be iterated using the fact that $J^\mu(x)$ is uncharged with respect to $Q[\Sigma^{(d-1)}]$,[4] resulting in

$$\left\langle Q^n[\Sigma^{(d-1)}]\mathcal{O}_1(x_1)\dots\mathcal{O}_k(x_k)\right\rangle = \chi^n(\Sigma^{(d-1)})\left\langle\mathcal{O}_1(x_1)\dots\mathcal{O}_k(x_k)\right\rangle. \tag{10}$$

This implies that the exponentiated operators (7) satisfy

$$\left\langle U_g[\Sigma^{(d-1)}]\mathcal{O}_1(x_1)\dots\mathcal{O}_k(x_k)\right\rangle = e^{i\alpha\chi(\Sigma^{(d-1)})}\left\langle\mathcal{O}_1(x_1)\dots\mathcal{O}_k(x_k)\right\rangle, \quad g = e^{i\alpha}. \tag{11}$$

More generally the integrated Ward identities associated to a finite transformation $g \in G$ can be written as

$$\left\langle U_g[\Sigma^{(d-1)}]\mathcal{O}_1(x_1)\dots\mathcal{O}_k(x_k)\right\rangle = \left\langle\mathcal{O}'_1(x_1)U_g[\Sigma'_{d-1}]\mathcal{O}_2(x_2)\dots\mathcal{O}_k(x_k)\right\rangle, \tag{12}$$

where $\mathcal{O}'_1(x_1) = (R_1(g)\cdot\mathcal{O}_1)(x_1)$ is the transformed operator according to its representation $R_1$ under $G$, $\Sigma^{(d-1)}$ is a surface linking with the point $x_1$ and $\Sigma'^{(d-1)}$ is its deformation across the point. The selection rules on correlation functions now follow from the fact that a topological operator $U_g[\Sigma^{(d-1)}]$ supported on a very big surface at infinity is trivial, but shrinking it to a point, $U_g$ passes and transforms all the local operators. We then get

$$\langle\mathcal{O}_1(x_1)\dots\mathcal{O}_n(x_n)\rangle = R_1(g)\dots R_n(g)\cdot\langle\mathcal{O}_1(x_1)\dots\mathcal{O}_n(x_n)\rangle, \tag{13}$$

which is the desired selection rule. A correlation function of local operators can be non-vanishing only if the direct product of representations contains the singlet representation. While $Q[\Sigma^{(d-1)}]$ and $U_g[\Sigma^{(d-1)}]$ enforce equivalent constraints on the theory, the advantage of using the exponentiated operator $U_g[\Sigma^{(d-1)}]$ is that in (12) we do not need to define the infinitesimal transformation $\delta\mathcal{O}$ so that the generalization to discrete symmetries is straightforward.

---

[3]In the following we suppress the group and algebra indices. In (7) the element $g \in G$ is the exponential of $\alpha$ valued in the dual of the Lie algebra of $G$.

[4]This is not true for non-abelian $G$. However with simple manipulations one can reach the same conclusion. Here we focus on the abelian case just for notational simplicity.

If we add a deformation of the pure theory which explicitly breaks $G$, the Ward identities (6) acquire a new term and, as expected, the operator $Q[\Sigma^{(d-1)}]$ (or equivalently $U_g[\Sigma^{(d-1)}]$) is no longer topological. For example, for $G = U(1)$ and a deformation described by the action (the term $h\mathcal{O}_0(x)$ is always paired with its hermitian conjugate, which we leave implicit)

$$S = S_0 + h \int d^d x \, \mathcal{O}_0(x), \tag{14}$$

where $\mathcal{O}_0(x)$ is a local operator with charge $q_0$ under $U(1)$ and $h$ is a coupling, we get

$$i\langle \partial_\mu J^\mu(x)\mathcal{O}_1(x_1)\dots\mathcal{O}_k(x_k)\rangle = \sum_{l=1}^{k} \delta^{(d)}(x - x_l)\langle \mathcal{O}_1(x_1)\dots\delta\mathcal{O}_l(x_l)\dots\mathcal{O}_k(x_k)\rangle \\ - ihq_0\langle \mathcal{O}_1(x_1)\dots\mathcal{O}_k(x_k)\mathcal{O}_0(x)\rangle. \tag{15}$$

Integrating over an open region $D^{(d)}$ with boundary $\Sigma^{(d-1)}$ we have

$$\Big\langle Q[\Sigma^{(d-1)}]\mathcal{O}_1\dots\mathcal{O}_k\Big\rangle = \chi(\Sigma^{(d-1)})\langle \mathcal{O}_1\dots\mathcal{O}_k\rangle - hq_0 \int_{D^{(d)}} d^d x \, \Big\langle \mathcal{O}_1\dots\mathcal{O}_k\mathcal{O}_0(x)\Big\rangle. \tag{16}$$

If the coupling $h$ is irrelevant, at large distances and for sufficiently large surfaces $\Sigma^{(d-1)}$, the second term in the r.h.s. of (16) is expected to be suppressed with respect to the first one, and the operators $Q(\Sigma^{(d-1)})$ become approximately topological.[5] In this case we say that the symmetry $G$ is *emergent* in the IR. For a related discussion on approximate symmetries in the language of topological operators see [64].

## 2.2 Quenched disorder and Ward identities

Theories with quenched disorder in the continuum limit can often be described starting from a pure theory $S_0$ and adding a perturbation like in (14) (see e.g. [65,66] for a modern analysis in this context), where $h$ is taken to be space-dependent (again we always implicitly pair up $h(x)\mathcal{O}_0(x)$ with its hermitian conjugate):

$$S[h] = S_0 + \int d^d x \, h(x)\mathcal{O}_0(x). \tag{17}$$

The random coupling is sampled from a distribution $P[h]$ and we should think of an ensemble of systems, each member being described by the action (17). Note that the considerations above on the explicit breaking are valid, with minor modifications, for each member of the ensemble.

A relevant example (which we will extensively consider in the sections 3 and 4) is the case of white noise, where $P[h]$ is Gaussian

$$P[h] \propto \exp\left(-\frac{1}{2v} \int d^d x \, h^2(x)\right), \tag{18}$$

parametrized by a coupling $v$ which governs the width of the Gaussian distribution. Dimensional analysis fixes the dimension of $v$ to be

$$[v] = d - 2\Delta_{\mathcal{O}_0}, \tag{19}$$

---

[5]Needless to say, when written in terms of bare operators, (16) is generally UV divergent. In particular, the second term in the right hand side requires also the renormalization of the operator $\mathcal{O}_0$.

where $\Delta_{\mathcal{O}_0}$ is the classical scaling dimension of the operator $\mathcal{O}_0$. The disorder is classically irrelevant in the RG sense when

$$\Delta_{\mathcal{O}_0} > \frac{d}{2}. \tag{20}$$

The equation (20) is called Harris criterion [67]. If the disorder is classically relevant or marginal, it has an important effect on the IR dynamics. For instance, other fixed points could emerge, so called random fixed points, which can also have logarithmic behavior (see section 4), or we could have no fixed points at all. When (20) is satisfied and disorder is weak, the IR behaviour of the system is unaffected by the impurities.[6]

Like in the pure theory case, if the coupling $h(x)$ breaks a symmetry $G$ and is irrelevant, then the symmetry $G$ will appear as an emergent symmetry in the IR theory. On the other hand, in disordered theories symmetries might also appear on average, but *exactly*, namely at all energy scales, independently on the scaling dimension of $h(x)$. It is important to keep into account this distinction in the considerations that will follow. The latter case is the one that we will call *disordered symmetries*.

The observables we are interested in are averaged correlation functions of local operators defined as (we adopt here the notation of [66])

$$\overline{\langle \mathcal{O}_1(x_1)\dots\mathcal{O}_k(x_k)\rangle} = \int \mathcal{D}h\, P[h] \frac{\int \mathcal{D}\mu\, e^{-S[h]} \mathcal{O}_1(x_1)\dots\mathcal{O}_k(x_k)}{\int \mathcal{D}\mu\, e^{-S[h]}}\,, \tag{21}$$

where $\mu$ is the path integral measure and $P[h]$ is an arbitrary distribution, not necessarily of the form (18). Correlation functions can be obtained as usual by coupling each local operator $\mathcal{O}_i$ to an external source $K_i$ and by taking functional derivatives with respect to the $K_i$'s of the averaged generating functional $Z_D[K_i]$ defined as

$$Z_D[K_i] := \int \mathcal{D}h\, P[h] \frac{\int \mathcal{D}\mu\, e^{-S[h]+\int K_i \mathcal{O}_i}}{\int \mathcal{D}\mu\, e^{-S[h]}} = \int \mathcal{D}h\, P[h] \frac{Z[K_i, h]}{Z[0, h]}\,. \tag{22}$$

We can also define the disordered free energy $W_D[K_i]$ as

$$W_D[K_i] := \int \mathcal{D}h\, P[h] \log Z[K_i, h] = \int \mathcal{D}h\, P[h] W[K_i, h] = \overline{W[K_i, h]}\,, \tag{23}$$

that generates averages of *connected* correlation functions

$$\overline{\langle \mathcal{O}_1(x_1)\dots\mathcal{O}_k(x_k)\rangle}_c = \frac{\delta^k W_D[K_i]}{\delta K_1(x_1)\dots\delta K_k(x_k)}\bigg|_{K_i=0}\,. \tag{24}$$

We stress that, unlike standard QFTs, in quenched disorder theories not all correlators can be determined from the connected ones and in particular

$$\overline{\langle \mathcal{O}_i(x)\rangle\langle \mathcal{O}_j(y)\rangle} \neq \overline{\langle \mathcal{O}_i(x)\rangle}\,\, \overline{\langle \mathcal{O}_j(y)\rangle}\,. \tag{25}$$

This is one of the crucial properties of disordered systems which will play an important role in the following. This motivates to introduce a more general generating functional

$$Z_D^{(N)}\Big[K_i^{(1)},\dots,K_i^{(N)}\Big] := \int \mathcal{D}h\, P[h] \prod_{j=1}^{N} \frac{Z\Big[K_i^{(j)}, h\Big]}{Z[0, h]}\,, \tag{26}$$

---

[6]When instead disorder is strong other fixed points may emerge, a standard example being the Nishimori fixed point in the 2d Random field Ising model, see e.g. [68].

whose functional derivatives produce the average of products of correlators. The generalization of (25) is

$$Z_D^{(N)}\left[K_i^{(1)},...,K_i^{(N)}\right] \neq \prod_{j=1}^N Z_D\left[K_i^{(j)}\right]. \tag{27}$$

Now suppose that the pure theory $S_0$ has some global 0-form invertible symmetry $G$. If the random deformation is $G-$invariant every realization of the system enjoys the symmetry, therefore $G$ is a symmetry of the full disordered theory and it will show up in the averaged correlators. Indeed from the Ward identities of the theory in presence of a random source $h(x)$, by simply taking the average we immediately get the expected identities. This applies also to higher-form symmetries which cannot be broken by adding local operators to the action [1,3].

If the random deformation breaks some or all of the symmetries of the pure theory, the story is more interesting. In this case we want to understand if and under which conditions the disordered theory still enjoys these symmetries. We start by considering an internal invertible continuous 0-form symmetry $G$, but our conclusions apply also in more general setups. In order to gain some intuition it is useful to use a spurionic argument. The path integral of the theory coupled to a random source $h(x)$ is

$$Z[h] = \int \mathcal{D}\mu \exp\left(-S_0 - \int h(x)\mathcal{O}_0(x)\right). \tag{28}$$

Because of the explicit breaking the partition function obeys

$$Z[h] = Z[R_0^\vee(g) \cdot h], \quad g \in G, \tag{29}$$

where $\mathcal{O}_0$ transforms in representation $R_0$ of $G$, and $R_0^\vee$ is its transpose. Turning on sources $K_i$ for operators of the pure theory we see that the generating functional satisfies

$$\begin{aligned}
Z[K_i,h] &= \int \mathcal{D}\mu \exp\left(-S_0 - \int h(x)\mathcal{O}_0(x) + \int K_i(x)\mathcal{O}_i(x)\right) \\
&= Z[R_i^\vee(g) \cdot K_i, R_0^\vee(g) \cdot h],
\end{aligned} \tag{30}$$

where we are assuming that $\mathcal{O}_i$ transform in representation $R_i$ of $G$ with $R_i^\vee$ being the transpose. Thus the correlators before averaging are not $G-$invariant but

$$\left.\frac{\delta}{\delta K_1 \dots \delta K_n} Z[K_i,h]\right|_{K_i=0} = R_1(g)\dots R_n(g) \cdot \left.\frac{\delta}{\delta K_1 \dots \delta K_n} Z[K_i, R_0^\vee(g) \cdot h]\right|_{K_i=0}. \tag{31}$$

This implies that

$$\begin{aligned}
\overline{\langle \mathcal{O}_1(x_1)\dots\mathcal{O}_n(x_n)\rangle} &= \int \mathcal{D}h\, P[h] \frac{1}{Z[h]} \left.\frac{\delta Z[K_i,h]}{\delta K_1 \dots \delta K_n}\right|_{K_i=0} \\
&= R_1(g)\dots R_n(g) \cdot \int \mathcal{D}h\, P[h] \frac{1}{Z[h]} \left.\frac{\delta Z[K_i, R_0(g)^\vee \cdot h]}{\delta K_1 \dots \delta K_n}\right|_{K_i=0}.
\end{aligned} \tag{32}$$

We can now change variable in the $h$-path integral, $R_0(g^{-1})^\vee \cdot h(x) \to h(x)$. Crucially, *if the probability measure $\mathcal{D}h\, P[h]$ is invariant*, the averaged correlator obeys the $G$ selection rules

$$\overline{\langle \mathcal{O}_1(x_1)\dots\mathcal{O}_n(x_n)\rangle} = R_1(g)\dots R_n(g) \cdot \overline{\langle \mathcal{O}_1(x_1)\dots\mathcal{O}_n(x_n)\rangle}, \tag{33}$$

but only on average. For example, a space-dependent coupling breaks translations, but if $P[h]$ is translation-invariant (like e.g. in (18)), then momentum conservation is recovered on average.

Although the above spurion analysis is enough to determine selection rules, it does not provide the explicit form of the conserved currents and which Ward identities are satisfied (and how). The existence of topological operators is not even guaranteed and the common lore which identifies symmetries with topological defects needs a more detailed analysis in order to be verified. Let us then derive the form of Ward identities for disordered symmetries. For notational simplicity we focus on $G = U(1)$, but the analysis can be extended to any Lie group. Consider the generating functional $Z_D[K_i]$ defined in (22). The usual Ward identities are derived by changing variables in the path integral at the numerator, transforming all the fields with a space-time dependent $U(1)$ element $e^{i\epsilon(x)}$, so that

$$S_0 \to S_0 + i \int \epsilon(x) \partial_\mu J^\mu(x), \tag{34}$$

$J^\mu$ being the Noether current. Here the symmetry is broken by $h(x)$ in any specific realization, nevertheless we can modify the standard procedure by changing variable also in the path integral at the denominator. Since $h(x)$ is space dependent, Poincaré invariance is explicitly broken in each specific realization and generally $\langle J^\mu \rangle \neq 0$. This suggests that even if the symmetry is recovered on average the current must be modified somehow. The above-mentioned change of variable in both numerator and denominator, expanding at first order in $\epsilon(x)$ leads to

$$\int \mathcal{D}h\, P[h] \left( \left\langle -\partial_\mu J^\mu - q_0 h \mathcal{O}_0 + q_i K_i \mathcal{O}_i \right\rangle_K + \frac{Z[K_i, h]}{Z[0, h]} \left\langle \partial_\mu J^\mu + q_0 h \mathcal{O}_0 \right\rangle \right) = 0. \tag{35}$$

Here $\langle \cdots \rangle_K$ represents correlators in the presence of the external sources $K_i$ whereas $\langle \cdots \rangle$ implicitly assumes that these sources are turned off. By taking functional derivatives with respect to the sources $K_i$ and then setting $K_i = 0$ we get

$$\overline{\left\langle \partial_\mu \widetilde{J}^\mu(x) \mathcal{O}_1(x_1) \cdots \right\rangle} + q_0 \overline{\left\langle h(x) \widetilde{\mathcal{O}}_0(x) \mathcal{O}_1(x_1) \cdots \right\rangle} = \sum_i q_i \delta^{(d)}(x - x_i) \overline{\left\langle \mathcal{O}_1(x_1) \cdots \right\rangle}, \tag{36}$$

where we introduced the shifted operators

$$\widetilde{J}^\mu(x; h(x)) := J^\mu(x) - \langle J^\mu(x) \rangle, \qquad \widetilde{\mathcal{O}}_0(x; h(x)) := \mathcal{O}_0(x) - \langle \mathcal{O}_0(x) \rangle. \tag{37}$$

The vacuum expectation values should be thought of as certain (generally non-local) functionals of $h(x)$, whose presence is important in the average.

Since $h(x)$ is integrated over all space-dependent configurations, the second term in (36) vanishes identically provided that the probability measure satisfies certain invariance conditions. Indeed we are allowed to perform the change of variable $h(x) \to e^{-iq_0\epsilon(x)}h(x)$ in the $h$ path integral of (22), and if the probability measure is invariant under this formal transformation we obtain

$$q_0 \int \mathcal{D}h P[h] \left( \left\langle h \mathcal{O}_0 \right\rangle_{K_i} - \frac{Z[K_i, h]}{Z[0, h]} \left\langle h \mathcal{O}_0 \right\rangle \right) = 0. \tag{38}$$

Taking arbitrary functional derivatives with respect to the external sources and setting them to zero we find

$$q_0 \overline{\left\langle h(x) \widetilde{\mathcal{O}}_0(x; h(x)) \mathcal{O}_1(x_1) \ldots \right\rangle} = 0, \tag{39}$$

which implies the vanishing of the second term in the left hand side of (36). From (15) with no $\mathcal{O}_1 \cdots \mathcal{O}_k$ insertions, we also get the relation

$$\left\langle \partial^\mu J_\mu(x) + q_0 h(x) \mathcal{O}_0(x) \right\rangle = 0, \tag{40}$$

valid before averaging. We are now ready to discuss Ward identities. If $q_0 = 0$, namely the $U(1)$ symmetry is unbroken in any realization of the ensemble, plugging (40) in (36) leads

to the averaged version of the ordinary Ward identities (6). This is of course expected, given that (6) holds even before average in this case. More interestingly, for $q_0 \neq 0$, thanks to (39) we find the disordered Ward identities

$$i\overline{\langle \partial_\mu \widetilde{J}^\mu(x; h(x))\mathcal{O}_1(x_1)\cdots \mathcal{O}_k(x_k)\rangle} = \sum_{i=1}^{k} iq_i \delta^{(d)}(x - x_i)\overline{\langle \mathcal{O}_1(x_1)\cdots \mathcal{O}_k(x_k)\rangle}. \qquad (41)$$

Several comments are in order.

- The relation we obtained has the same form of a standard Ward identity, but for a modified current $\widetilde{J}^\mu = J^\mu - \langle J^\mu \rangle$. The modification is proportional to the identity operator in any of the specific realization of the ensemble, and can be thought of as an $h-$dependent counterterm which restores the conservation in the disordered theory. Note that the Ward identities written as in (41) apply for arbitrary correlation functions of local operators which *do not* contain explicit powers of $h(x)$.

- Before averaging the current $J^\mu$ (as well as its shifted version $\widetilde{J}^\mu$) is sensitive to the UV renormalization of the theory, i.e. it acquires a non-vanishing anomalous dimension (in contrast to ordinary conserved currents in pure theories). A proper definition of $J^\mu$ would require a regularization of the theory and a choice of renormalization scheme. Luckily enough, if we are *only* interested in averaged correlators, we do not need to worry about these issues, since (41) guarantees that $\widetilde{J}^\mu$ is effectively conserved inside averaged correlators.

- The Ward identities (41) are valid independently of the behavior of the current at infinity. When the integral of $\partial_\mu \widetilde{J}^\mu$ over the full space diverges (this requires the space to be non compact) the disordered symmetry is spontaneously broken. We do not discuss spontaneous disordered symmetry breaking in detail in this paper. We briefly comment on it in the conclusions. If the symmetry is not spontaneously broken the integral of $\partial_\mu \widetilde{J}^\mu$ over the full space vanishes. Thus (41) implies the selection rules we already derived from the spurionic argument. However (41) is a more refined constraint being a local conservation equation: local currents can be used to discuss 't Hooft anomalies and eventually gauging the symmetry, as we will see shortly. Moreover we will show in the next subsection that, with some modification with respect to the usual story, the conservation of $\widetilde{J}^\mu$ leads to topological operators as in the pure case.

- Since the random coupling $h(x)$ is space dependent, in every member of the ensemble translational symmetry is explicitly broken. The analysis above can be repeated for the stress-energy tensor $T^{\mu\nu}$, showing that also traslational invariance is recovered in a theory with quenched disorder, provided $P[h]$ is translational invariant. Similar considerations apply for rotations.

With simple modifications we have a similar identity for any Lie group $G$:

$$i\overline{\langle \partial_\mu \widetilde{J}_a^\mu(x; h(x))\mathcal{O}_1(x_1)\cdots\rangle} = \sum_i \delta^{(d)}(x - x_i)\overline{\langle \mathcal{O}_1(x_1)\cdots r_i(T_a)\cdot \mathcal{O}_i(x_i)\cdots\rangle}. \qquad (42)$$

Here $T_a$ is a Lie algebra generator and $r_i$ is the representation of the Lie algebra, induced by $R_i$, under which $\mathcal{O}_i$ transforms. A more general situation could take place, in which the disorder deformation does not break the full group, but leaves a subgroup $H \subset G$ unbroken. In this case any specific realization is $H-$symmetric, and thus the currents $J_\alpha^\mu$, with $T_\alpha$ generator of $\mathfrak{h} = \mathrm{Lie}(H)$, satisfy the standard Ward identity without the necessity of averaging. In particular $\langle \partial_\mu J_\alpha^\mu \rangle = 0$, even if the expectation value of the current itself is not necessarily vanishing due

to the lack of Poincaré invariance. Even if $G/H$ is generically not a group, the associated currents, which are not conserved in any specific realization, after the appropriate shift by their expectation values turn out to satisfy the Ward identity (42) in the disordered theory, and reconstruct the full group $G$.

Sometimes a 0-form symmetry $G$ can form a higher-group structure with higher-form symmetries of the theory [4–6]. In this case $G$ is not really a subgroup of the full symmetry structure, since the product of several $G$—elements can also produce an element of the higher-form symmetry. This kind of extension is classified by group-cohomology classes, the Postnikov classes: for instance in a 2-group, mixing $G$ with a 1-form symmetry $\Gamma$, the relevant datum is a class $\beta \in H^3(BG, \Gamma)$, with $BG$ the classifying space of $G$. The important thing is that this is a discrete datum and cannot change under continuous deformation. Suppose we add a disorder breaking $G$, and this re-emerges as a disordered symmetry. A natural question is whether the higher-group structure is also recovered. The answer is affirmative as a consequence of the discrete nature of this structure. Indeed the probability distributions $P[h]$ have some tunable continuous parameters, like $v$ in the Gaussian case (18), such that the pure theory is recovered in some limit ($v \to 0$ in the Gaussian case). The cohomology class characterising the higher-group is discrete and cannot change with this continuous parameter. Since all these disordered theories are continuously connected to the pure one, the higher-group structure is unchanged.

Up to this point disordered symmetries seem to behave like ordinary global symmetries in pure theories. The difference arises by considering averages of products of correlators

$$\overline{\prod_{j=1}^{N} \left\langle \mathcal{O}_1^{(j)}(x_1^{(j)}) \cdots \mathcal{O}_{k_j}^{(j)}(x_{k_j}^{(j)}) \right\rangle}. \tag{43}$$

Because of (27) these are independent correlators, and we do not expect them to satisfy Ward identities immediately implied by (41), or to be constrained by the usual selection rules. Let us consider the more general generating functional $Z_D^{(N)}[\{K_i^{(j)}\}]$ introduced in (26). With the same manipulations which led to (38), we get

$$q_0 \sum_{j=1}^{N} \int \mathcal{D}h P[h] \left( \left( \langle h\mathcal{O}_0 \rangle_{K_i^{(j)}} - \frac{Z[K_i^{(j)}, h]}{Z[0, h]} \langle h\mathcal{O}_0 \rangle \right) \prod_{l \neq j} \frac{Z[K_i^{(l)}, h]}{Z[0, h]} \right) = 0, \tag{44}$$

while the individual terms of the sum are generically non-vanishing. This implies that the only Ward identity we can prove from $Z_D^{(N)}[\{K_i^{(j)}\}]$ are obtained by changing variable in *all* the path integrals involved: if we try to change variables only in a subset of these path integrals, the extra term arising would be not be of the form (44), but the sum would be over that subset of indices. Repeating the steps above we obtain the Ward identities for averages of products of correlators:

$$\sum_{j=1}^{N} \overline{\left\langle \partial_\mu \tilde{J}^\mu \mathcal{O}_1^{(j)} \cdots \mathcal{O}_{k_j}^{(j)} \right\rangle \left( \prod_{l \neq j} \left\langle \mathcal{O}_1^{(l)} \cdots \mathcal{O}_{k_l}^{(l)} \right\rangle \right)} = \sum_{j=1}^{N} \sum_{i_j=1}^{k_j} q_{i_j}^{(j)} \delta^{(d)}(x - x_{i_j}^{(j)}) \overline{\prod_{l=1}^{N} \left\langle \mathcal{O}_1^{(l)} \cdots \mathcal{O}_{k_l}^{(l)} \right\rangle}. \tag{45}$$

These Ward identities imply weaker selection rules. For instance, the correlator

$$\overline{\left\langle \mathcal{O}_1(x_1) \cdots \mathcal{O}_{k_1}(x_{k_1}) \right\rangle \left\langle \mathcal{O}_{k_1+1}(x_{k_1+1}) \cdots \mathcal{O}_{k_1+k_2}(x_{k_1+k_2}) \right\rangle}, \tag{46}$$

can be non zero when $\sum_{i=1}^{k_1} q_i \neq 0$ and $\sum_{i=k_1+1}^{k_1+k_2} q_i \neq 0$, provided that $\sum_{i=1}^{k_1+k_2} q_i = 0$.

In a theory with quenched disorder ordinary and disordered symmetries can be present at the same time, and we see that their different action shows up in looking at averages of products of correlators.

See appendix A.1.1 for an explicit derivation of (41) for a two-point ($k = 2$) function in a simple solvable model.

## 2.3 Topological operators for disordered symmetries

We now address the question of whether there exist topological symmetry operators implementing disordered symmetries, placing them in the general framework of [1]. This is important to e.g. generalize to discrete symmetries, coupling them to backgrounds and discuss non-perturbative anomalies. For notational simplicity we again focus on the $G = U(1)$ case, but all the considerations can be extended to any Lie group. We introduce the modified charge operator

$$\widetilde{Q}[\Sigma^{(d-1)}; h(x)] = \int_{\Sigma^{(d-1)}} \widetilde{J}_\mu n^\mu = Q[\Sigma^{(d-1)}] - \langle Q[\Sigma^{(d-1)}] \rangle, \tag{47}$$

which satisfies the integrated Ward identity

$$\overline{\langle \widetilde{Q}[\Sigma^{(d-1)}; h] \mathcal{O}_1(x_1) \dots \mathcal{O}_k(x_k) \rangle} = \chi(\Sigma^{(d-1)}) \overline{\langle \mathcal{O}_1(x_1) \dots \mathcal{O}_k(x_k) \rangle}, \tag{48}$$

with $\chi(\Sigma^{(d-1)})$ as in (9), as well as the generalization to arbitrary products

$$\sum_{j=1}^{N} \overline{\langle \widetilde{Q}[\Sigma^{(d-1)}; h] \mathcal{O}_1^{(j)} \cdots \mathcal{O}_{k_j}^{(j)} \rangle \left( \prod_{l \neq j} \langle \mathcal{O}_1^{(l)} \cdots \mathcal{O}_{k_l}^{(l)} \rangle \right)} = \chi(\Sigma^{(d-1)}) \overline{\prod_{l=1}^{N} \langle \mathcal{O}_1^{(l)} \cdots \mathcal{O}_{k_l}^{(l)} \rangle}. \tag{49}$$

The reason why the naive procedure of constructing the symmetry operator by exponentiating $\widetilde{Q}[\Sigma^{(d-1)}]$ does not work can be already understood at the second order: $\widetilde{Q}^2[\Sigma^{(d-1)}]$ does not measure the square of the total charge. Let $\Phi$ be a generic product of local operators.[7] We have

$$\overline{\langle \widetilde{Q}^2 \Phi \rangle} = \overline{\langle \widetilde{Q} Q \Phi \rangle} - \overline{\langle Q \rangle \langle \widetilde{Q} \Phi \rangle} = \chi \overline{\langle Q \Phi \rangle} - \chi \overline{\langle Q \rangle \langle \Phi \rangle} + \overline{\langle \widetilde{Q} Q \rangle \langle \Phi \rangle} = \chi^2 \overline{\langle \Phi \rangle} + \overline{\langle \widetilde{Q} Q \rangle \langle \Phi \rangle}. \tag{50}$$

In the second step we used both the Ward identity (48) and (49) with $N = 2$. We deduce that what measures the total charge square is not $\widetilde{Q}^2$ but

$$\widetilde{Q}_2 := \widetilde{Q}^2 - \langle \widetilde{Q} Q \rangle = Q^2 - 2\langle Q \rangle Q + 2\langle Q \rangle^2 - \langle Q^2 \rangle. \tag{51}$$

In order to construct the topological symmetry operator we need, for any $n \in \mathbb{N}$, an operator $\widetilde{Q}_n$ such that

$$\overline{\langle \widetilde{Q}_n \mathcal{O}_1 \cdots \mathcal{O}_k \rangle} = \chi^n \overline{\langle \mathcal{O}_1 \cdots \mathcal{O}_k \rangle}, \tag{52}$$

and then define the symmetry operators as

$$\widetilde{U}_g = \sum_{n=0}^{\infty} \frac{(i\alpha)^n}{n!} \widetilde{Q}_n, \qquad g = e^{i\alpha}. \tag{53}$$

To prove that such operators exist, and show how to compute them, we start from $\overline{\langle Q^n \Phi \rangle}$ (again $\Phi$ denotes a generic product of local operators), and we rewrite one $Q$ as $\widetilde{Q} + \langle Q \rangle$, so that we can use a linear Ward identity for $\widetilde{Q}$, and we iterate until we eliminate all the $Q$s:

$$\begin{aligned}
\overline{\langle Q^n \Phi \rangle} &= \overline{\langle \widetilde{Q} Q^{n-1} \Phi \rangle} + \overline{\langle Q \rangle \langle Q^{n-1} \Phi \rangle} = \chi \overline{\langle Q^{n-1} \Phi \rangle} + \overline{\langle Q \rangle \langle Q^{n-1} \Phi \rangle} \\
&= \chi^2 \overline{\langle Q^{n-2} \Phi \rangle} + \chi \overline{\langle Q \rangle \langle Q^{n-2} \Phi \rangle} + \overline{\langle Q \rangle \langle Q^{n-1} \Phi \rangle} \\
&\ \ \vdots \\
&= \chi^n \overline{\langle \Phi \rangle} + \sum_{k=0}^{n-1} \chi^k \overline{\langle Q \rangle \langle Q^{n-k-1} \Phi \rangle}.
\end{aligned} \tag{54}$$

---

[7]In order to avoid cluttering in the formulas, from now on we will adopt a lighter notation omitting often the support of local operators or indices.

The terms $\chi^k \overline{\langle Q \rangle \langle Q^{n-k-1}\Phi \rangle}$ can be managed as follows. We eliminate one $\chi$ by using the linear Ward identity for the averaged product of two correlators for $\widetilde{Q}$, which we then re-expand as $Q - \langle Q \rangle$:

$$
\begin{aligned}
\chi^k \overline{\langle Q \rangle \langle Q^{n-k-1}\Phi \rangle} &= \chi^{k-1}\Big( \overline{\langle \widetilde{Q}Q \rangle \langle Q^{n-k-1}\Phi \rangle} + \overline{\langle Q \rangle \langle \widetilde{Q}Q^{n-k-1}\Phi \rangle} \Big) \\
&= \chi^{k-1}\Big( \overline{\langle Q^2 \rangle \langle Q^{n-k-1}\Phi \rangle} - 2\overline{\langle Q \rangle^2 \langle Q^{n-k-1}\Phi \rangle} + \overline{\langle Q \rangle \langle Q^{n-k}\Phi \rangle} \Big).
\end{aligned}
\tag{55}
$$

Then we eliminate another $\chi$ from each term, again using the linear Ward identity, in some terms with the product of two correlators, in others with the product of three correlators. We continue in this way until we eliminate all the $\chi$s, and remain with a sum of averages of products of expectation values of $\langle Q^a \rangle$ for various $a$, and $\langle Q^b \Phi \rangle$ for a certain $b$, generally different for each term. This defines the operator $\widetilde{Q}_n$. For instance

$$
\widetilde{Q}_3 = Q^3 - 3\langle Q \rangle Q^2 - 3\langle Q^2 \rangle Q + 6\langle Q \rangle^2 Q - \langle Q^3 \rangle + 6\langle Q \rangle \langle Q^2 \rangle - 6\langle Q \rangle^3 .
\tag{56}
$$

While this seems very complicated, one can check until arbitrarily high order that the expansion can be beautifully resummed as

$$
\widetilde{U}_g = \sum_{n=0}^{\infty} \frac{(i\alpha)^n}{n!} \widetilde{Q}_n = e^{i\alpha Q} \langle e^{i\alpha Q} \rangle^{-1} ,
\tag{57}
$$

where $\widetilde{Q}_0 := 1$. Note that this is the only result consistent with $\langle \widetilde{U}_g \rangle = 1$, which must be true by construction since $\langle \widetilde{Q}_n \rangle = 0$ as a direct consequence of the Ward identities (52) satisfied by $\widetilde{Q}_n$ in absence of local operators.

The operator $\widetilde{U}_g$ in averaged correlators behaves as

$$
\overline{\langle \widetilde{U}_g[\Sigma^{(d-1)}; h(x)] \mathcal{O}_1 \cdots \mathcal{O}_k \rangle} = e^{i\alpha \chi(\Sigma^{(d-1)})} \overline{\langle \mathcal{O}_1 \cdots \mathcal{O}_k \rangle},
\tag{58}
$$

and is hence a *topological symmetry operator, on average*. It satisfies the group law

$$
\overline{\langle \widetilde{U}_g \widetilde{U}_h \Phi \rangle} = \overline{\langle \widetilde{U}_{gh} \Phi \rangle},
\tag{59}
$$

$\Phi$ being an arbitrary product of local operators. As a consequence, the naive expectation that $e^{i\alpha Q} e^{i\beta Q} = e^{i(\alpha+\beta)Q}$ is wrong because of the disorder. Note that before averaging the operator $\widetilde{U}$ is subject to renormalization and its proper definition requires a choice of renormalization scheme. We do not need to keep track of these subtleties, however, because they are washed away after the average is taken.

We now consider how $\widetilde{U}_g$ behaves inside averages of products of correlators (43), extending (49) to finite symmetry actions. This is important because, as we mentioned, products of correlators is what really characterizes disordered symmetries with respect to ordinary ones, and we need the symmetry operator version of the criterion we discussed at the end of section 2.2. In principle one could explicitly construct the correct combination of charges $\widetilde{Q}_n$ entering the Ward identities using the results above. For example, in the average of products of two correlators, at quadratic order in the charges we have

$$
\overline{\langle \widetilde{Q}_2 \Phi_1 \rangle \langle \Phi_2 \rangle} + \overline{\langle \Phi_1 \rangle \langle \widetilde{Q}_2 \Phi_2 \rangle} + 2\overline{\langle \widetilde{Q}_1 \Phi_1 \rangle \langle \widetilde{Q}_1 \Phi_2 \rangle} = \chi^2 \overline{\langle \Phi_1 \rangle \langle \Phi_2 \rangle},
\tag{60}
$$

$\Phi_{1,2}$ being two distinct generic products of local operators. Similarly for multiple products.

We claim that the correct Ward identities consist in inserting $\widetilde{U}_g$ in all the (un)factorized correlators under average:

$$
\overline{\prod_{j=1}^{N} \langle \widetilde{U}_g[\Sigma^{(d-1)}; h(x)] \mathcal{O}_1^{(j)}(x_1^{(j)}) \cdots \mathcal{O}_{k_j}^{(j)}(x_{k_j}^{(j)}) \rangle} = e^{i\alpha \chi(\Sigma^{(d-1)})} \overline{\prod_{j=1}^{N} \langle \mathcal{O}_1^{(j)}(x_1^{(j)}) \cdots \mathcal{O}_{k_j}^{(j)}(x_{k_j}^{(j)}) \rangle}.
\tag{61}
$$

This can be checked by expanding both members in powers of $\alpha$, which gives a series of Ward identities for the $\widetilde{Q}_n$'s. For example, for two correlators ($N = 2$) we have

$$\sum_{l=0}^{k} \binom{k}{l} \overline{\left\langle \widetilde{Q}_l \Phi_1 \right\rangle \left\langle \widetilde{Q}_{k-l} \Phi_2 \right\rangle} = \chi^k \overline{\left\langle \Phi_1 \right\rangle \left\langle \Phi_2 \right\rangle}, \tag{62}$$

where $\chi = \chi_1 + \chi_2$ are the sum of charges of the local operators in the product $\Phi_{1,2}$ which are inside the support of the charge operators. Checking (62) directly is cumbersome, but we can proceed as follows. We rewrite the last term appearing in (54) using (62) (assuming its validity) with $\Phi_1 = Q$ and $\Phi_2 = Q^{n-k-1}\Phi$.[8] In this way we get

$$\widetilde{Q}_n = Q^n - \sum_{k=0}^{n-1} \sum_{l=0}^{k} \binom{k}{l} \langle \widetilde{Q}_l Q \rangle \widetilde{Q}_{k-l} Q^{n-k-1} = Q \widetilde{Q}_{n-1} - \sum_{l=0}^{n-1} \binom{n-1}{l} \langle \widetilde{Q}_l Q \rangle \widetilde{Q}_{n-l-1}. \tag{63}$$

This is a recursion formula which determines $\widetilde{Q}_n$ in terms of all the $\widetilde{Q}_m$ for $m < n$, and it is equivalent to (62). It can be checked that computing the topological charges with this formula gives the same result as computing them directly from the linear Ward identities, proving in this way the validity of (61) and (62).

For averages of multiple correlators the group law (59) generalizes to

$$\overline{\prod_{j=1}^{N} \left\langle \widetilde{U}_g \widetilde{U}_h \Phi_j \right\rangle} = \overline{\prod_{j=1}^{N} \left\langle \widetilde{U}_{gh} \Phi_j \right\rangle}. \tag{64}$$

We are finally able to characterize disordered symmetries in full generality. These are symmetries of theories with quenched disorder implemented by symmetry operators $\widetilde{U}_g$, $g \in G$, which become topological after quenched average. They satisfy the identity (58) and the group law (59) as operator equations valid in any averaged correlator. Differently from ordinary global symmetries, in averages of products of correlators like (43) they are topological only if inserted in each factor of the product, and satisfy the generalized group law (64) inside averaged correlators. Disordered symmetries are symmetries of the pure system broken by the disorder but with a symmetric probability measure. It is then not surprising that $\widetilde{U}_g$ can be written in terms of the corresponding topological operator $U_g$ of the pure system as

$$\widetilde{U}_g \left[ \Sigma^{(d-1)}; h(x) \right] = U_g \left[ \Sigma^{(d-1)} \right] \left\langle U_g \left[ \Sigma^{(d-1)} \right] \right\rangle^{-1}. \tag{65}$$

However the characterization above is intrinsic and does not require to know the pure system. The resummation of the series (53) into the compact expression (65) allows us to immediately generalize the analysis to more general groups $G$, including discrete ones where there is no current or charge operator available.

## 2.4  't Hooft anomalies for continuous disordered symmetries

We examine in this and the next subsections some general properties of disordered symmetries. We will argue that the concept of 't Hooft anomalies, for both continuous and discrete symmetries, extends to this context. In particular we show that disordered symmetries inherit the anomaly of their pure counterpart. This is important because we can use anomalies to constraint the IR dynamics of quenched disordered theories, whose flow is generally extremely

---

[8]Note that $\Phi$ can include integrated current operators $J_\mu$, hence powers of charges $Q$, but *not* powers of $\widetilde{Q}$. The latter is still the integral of a local operator, but with an explicit dependence on $h(x)$, in which case the analysis does not apply.

complicated. We start discussing 't Hooft anomalies for continuous disordered symmetries, postponing to section 2.5 the case of discrete symmetries.

A theory with a global symmetry can be coupled to a background gauge field $A$ which acts as an external source for the conserved current $J$, and results in a partition function $Z[A]$. A 't Hooft anomaly arises whenever $Z[A]$ is not invariant under gauge transformations of the background (see e.g. [69] for a modern review). Denoting by $A^\lambda$ the gauge transformed background with gauge parameter $\lambda$, we have

$$Z[A^\lambda] = e^{i \int_{X^{(d)}} \alpha(\lambda, A)} Z[A], \tag{66}$$

where the phase $\alpha(\lambda, A)$ in the exponent is the t'Hooft anomaly, a functional depending on $\lambda$ and $A$, which cannot be cancelled by local counterterms. Coupling to backgrounds for disordered symmetries is more subtle, because the symmetry is explicitly broken in any specific realization of the ensemble. If the symmetry is restored on average, however, a coupling to an external background becomes possible via the shifted current $\widetilde{J}$ defined in (37), namely we define

$$\overline{Z[A]} = \int \mathcal{D}h\, P[h] \int \mathcal{D}\mu e^{-S_0 - \int h(x)\mathcal{O}_0(x) + \int A_\mu \widetilde{J}^\mu}. \tag{67}$$

An 't Hooft anomaly for a disordered symmetry $G$ can be defined in close analogy with the ordinary case (66):

$$\overline{Z[A^\lambda]} = e^{i \int_{X^{(d)}} \alpha(\lambda, A)} \overline{Z[A]}. \tag{68}$$

Anomalies (both continuous and discrete) are known to be invariant under RG flow thanks to their topological nature (typically associated to a Chern-Simons level taking value in a cohomology group, see e.g. [70, 71]). In particular, the value of the anomaly cannot depend on possible continuous parameters entering in the disorder distribution $P[h]$, such as $v$ in the Gaussian example (18). By adiabatically changing such parameters, we can make the distribution arbitrarily peaked around $h = 0$, in which case we effectively recover the pure theory.[9] We then expect that a 't Hooft anomaly (68) associated to a disordered symmetry $G$ can only appear if the associated pure theory (before adding the disorder perturbation) had a 't Hooft anomaly for the same symmetry $G$. Moreover, the two anomalies must coincide. This can be easily verified for all anomalies which, from a path integral point of view, can be seen to derive from the non-invariance of the path integral measure [72]. Starting from the left hand side of (68) when $\lambda$ is infinitesimal, we perform a change of variable in the path integral in $Z[A^\lambda]$, which corresponds to an $x$-dependent transformation under $G$ such that $A^\lambda \to A$. As in pure theories, the non-invariance of the measure leads to the anomaly term. The derivative of the current coming from the action variation is cancelled by the explicit symmetry breaking term and we are left with the anomalous term only. Crucially, the latter does not depend on the disorder $h$ and hence we immediately get the infinitesimal version of the right hand side of (68), where $\alpha$ is exactly the same as in the underlying pure theory. If the anomaly vanishes, the disordered symmetry can be gauged by making the gauge field $A_\mu$ in (67) dynamical.

We report in appendix A.2 an example of matching of 't Hooft anomalies between the pure and the disorder theories using the replica trick, which will be introduced in section 3, for the case of the $U(1)$ chiral anomaly in four dimensions. Notice that the same example already provides an anomaly matching constraint in a disordered theory. Indeed a given member of the ensemble is clearly gapped while after the average the resulting disorder theory must be gapless due to the 't Hooft anomaly. Since the disorder in this case is classically irrelevant this is not surprising. A less trivial example can be found in the 2-dimensional version of the same model, where the same considerations apply but the disorder is classically marginal.

---

[9]For the gaussian case this is achieved by taking $v \to 0$.

## 2.5  Discrete disordered symmetries: 't Hooft anomalies and gauging

The topological operators $U_g[\Sigma^{(d-1)}]$ are crucial to handle discrete symmetries for which there is no current. In pure theories the coupling to background gauge fields associated to a discrete symmetry group $G$ can be achieved by modifying the path integral with the topological symmetry operators [1]. There are several equivalent ways to introduce a background gauge field for a discrete symmetry group $G$. One of these (see e.g. [5] for further details) consists in taking an atlas $\{U_i\}$ of the $d$-dimensional space $X^{(d)}$ and assigning group-valued connections $A_{ij} \in G$ on $U_i \cap U_j$ such that $A_{ij} = A_{ji}^{-1}$ and $A_{ij}A_{jk}A_{ki} = 1$ on triple intersections $U_i \cap U_j \cap U_k$. A codimension one symmetry operator $U_{g_p}[\Sigma_p^{(d-1)}]$ assigns $A_{ij} = g_p$ (or $g_p^{-1}$ depending on its orientation) if $\Sigma_p^{(d-1)}$ has a non trivial intersection number with the line dual to $U_i \cap U_j$ and $A_{ij} = 1$ otherwise.[10] The resulting sets of connections $A_{ij}$ defines a background gauge field for $G$ and can be represented by a cohomology class $A \in H^1(X^{(d)}, G)$. The operators $U_{g_p}[\Sigma_p^{(d-1)}]$ can intersect in three-valent junctions of codimension two provided that

$$g_i g_j g_k = 1\,, \tag{69}$$

or also in higher multi-valued junctions. The configuration described above requires few choices, and one must check independence on those. Since the operators are topological local changes in their support are immaterial. We could also change the mesh locally near the junctions, which corresponds to resolve a multi-valent junction in three-valent ones in different ways. This corresponds to background gauge transformations and a non-invariance under them signals a 't Hooft anomaly for discrete symmetries. In $d$ dimensions a 't Hooft anomaly is classified by a class $\alpha \in H^{d+1}(BG, U(1))$.[11]

Consider now a theory $T$ with quenched disorder, obtained by deforming a pure theory $T_0$, and denote by $T_h$ the member of the ensemble with coupling $h(x)$. Suppose $T$ has a discrete disordered symmetry $G$. As we have seen this is implemented by the operators

$$\widetilde{U}_g[\Sigma^{(d-1)}; h] = U_g[\Sigma^{(d-1)}]\Big\langle U_g[\Sigma^{(d-1)}]\Big\rangle^{-1}\,. \tag{70}$$

We introduce a fine-enough mesh of topological operators $\widetilde{U}_{g_i}[\Sigma_i^{(d-1)}; h]$ satisfying (on average) the cocycle condition (69) in the three-valent junctions. Since $\widetilde{U}_{g_i}[\Sigma_i^{(d-1)}; h]$ is not topological in $T_h$, the junctions (as well as the operators $\widetilde{U}$ themselves) are not really well-defined because of UV divergences. However we can employ an arbitrary regularization scheme for these divergences, without the need of specifying a renormalization scheme to try to define the junctions and the operators $\widetilde{U}$ (recall the second comment after (41)). This is because we know that the operators become topological after the average and hence such divergences are expected to be washed away from the integration over $h(x)$. We define

$$\begin{aligned}
Z_{T_h}\big[\{g_i\}, h(x)\big] &= \int \mathcal{D}\mu\, e^{-S[\phi] - \int h(x)\mathcal{O}_0(x)} \prod_i \widetilde{U}_{g_i}[\Sigma_i^{(d-1)}; h(x)] \\
&= \Big\langle \prod_i \widetilde{U}_{g_i}[\Sigma_i^{(d-1)}; h(x)]\Big\rangle\,,
\end{aligned} \tag{71}$$

which, contrary to the pure case, *does* depend on the specific location of the planes $\Sigma_i^{(d-1)}$. At this point there is no notion of background gauge fields. However, as a consequence of the

---

[10]In the dual triangulation the charts $U_i$ are points, the intersections $U_i \cap U_j$ are lines, and so on.

[11]Strictly speaking, this is the case for bosonic theories in $d < 3$ dimensions. More in general, anomalies are classified by a cobordism group [73].

Ward identity discussed in section 2.3,

$$Z_T\big[\{g_i\}\big] = \int \mathcal{D}h\, P[h]\, Z_{T_h}\big[\{g_i\}, h(x)\big], \tag{72}$$

is independent of the choice of location for $\Sigma_i$ and hence the set of operators $\widetilde{U}_{g_i}$ inserted in (72) corresponds to a well-defined discrete gauge field $A \in H^1(X^{(d)}, G)$. It is important to emphasize here that the gauge field $A$ arises only after the average over $h(x)$ is performed. Differently said, if a pure system has a symmetry $G$, perturbing it with quenched disorder and coupling it to a background are non-commutative operations. In what follows we denote the above partition function by $Z_T[A]$.

Local modifications of the three-valent junctions change the gauge field by an exact 1-cocycle $A \to A^\lambda = A + \delta\lambda$. This can change the partition function $Z_T[A]$ by a phase, which represents a class $\alpha \in H^{d+1}(BG, U(1))$: this is the diagnostic for an 't Hooft anomaly for a discrete disordered symmetry. Since the topological operator $\widetilde{U}_g[\Sigma^{(d+1)}; h(x)]$ is different from the one in the pure theory by the stacking of an $h(x)-$dependent functional, it is not a priori obvious that the contact terms arising in the local moves are the same as those in the pure theory, precisely as it occurred in the continuous case discussed in section 2.4. However, the fact that anomalies are classified by classes in $H^{d+1}(BG, U(1))$, which are discrete, immediately proves that they cannot depend on the strength of the disorder and must be equal to those of the pure theory. As a result, a system with a disordered symmetry with a 't Hooft anomaly cannot be trivially gapped. This is in agreement with previous works in condensed matter where – mostly in the context of topological insulators [48–52, 54] but not only (see e.g. [53]) – SPT phases of matter where the symmetry is disordered were found. We see that in general disordered symmetries can lead to protected non-trivial topological phases (see [55] for a recent analysis).[12]

Now suppose that the 't Hooft anomaly vanishes. Then $Z_T[A]$ is well defined and is possible to gauge the symmetry by summing over all consistent insertions of symmetry operators, or equivalently over cohomology classes $A \in H^1(X^{(d)}, G)$. We denote the resulting theory by $T/G$, whose partition function is[13]

$$Z_{T/G} = \sum_{A \in H^1(X^{(d)}, G)} Z_T[A]. \tag{73}$$

At this point everything is essentially the same as in the pure case (see e.g. [1, 7]). The operators of $T$ with a counterpart in $T/G$ are the gauge-invariant ones, while we also add the $(d-2)$ dimensional operators in the twisted sector of $G$. Indeed the topological operators $\widetilde{U}_g[\Sigma^{(d-1)}; h(x)]$ become trivial in $T/G$, and their boundary operators turn into genuine operators (on average). Finally, since $A \in H^1(X^{(d)}, G)$ is dynamical, $T/G$ has a dual symmetry generated by the Wilson lines of the $G$ gauge field. This is a $(d-2)$-form symmetry whose charged objects are the operators coming from the twisted sectors of $G$. For $G$ abelian the symmetry is the Pontryagin dual $G^\vee$, while it is a non-invertible symmetry in the non-abelian case [7].

---

[12]In [55] it is considered a Lorentizan theory with a disorder coupling depending on space but not on time. In this set-up it is found that purely disordered symmetries, i.e. in absence of pure symmetries, necessarily have a trivial t' Hooft anomaly. This is not in contradiction with our findings, based on Euclidean theories.

[13]In the pure case it is possible to modify this sum weighting the terms with phases. Consistency conditions related with associativity constraint these phases to be of the form $\int_{X^{(d)}} A^* \nu$, where $\nu \in H^d(BG, U(1))$ is a *discrete torsion class* and we think $A$ as a homotopy class of maps $X^{(d)} \to BG$, so that $A^* \nu \in H^d(X^{(d)}, U(1))$. Since the same kind of constraints are valid also in the disordered theories, we expect the very same modification of the gauging procedure to be possible also in this context.

# 3 Disordered symmetries and the replica trick

Disordered systems are often treated by means of the *replica trick*, which expresses the averaged correlation functions as certain limits of correlation functions of a standard QFT, the replica theory. In this section we interpret the disordered symmetries from the point of view of the replica theory. In addition to provide a sanity check of the results found in section 2, the method of replicas allows us to consider emergent symmetries in the disordered theory for which the results in the previous section do not apply. We will discuss emergent symmetries in section 4. For the rest of this section and the next section we assume a Gaussian probability distribution like (18) (and its generalization for complex $h$) with variance $v$.[14]

## 3.1 The replica trick

To fix our notation we briefly review the *replica trick*. This is a useful tool that allows to compute connected and full (i.e. both its connected and disconnected parts) correlators of the disordered theory as limits of correlators of a pure theory. The starting point of the replica trick is the identity

$$W = \log Z = \lim_{n \to 0} \left( \frac{\partial Z^n}{\partial n} \right). \tag{74}$$

The idea is to replicate the pure system $n$ times, indexing each copy with a label $a$

$$Z^n[h, K_i] = \int \prod_{a=1}^n \mathcal{D}\mu_a \exp\left( -\sum_a S_{0,a} - \sum_a \int h(x)\mathcal{O}_{0,a}(x) + \sum_{i,a} \int K_i(x)\mathcal{O}_{i,a}(x) \right), \tag{75}$$

with the same random field coupling $h$ and external sources $K_i$ for all replicas. When $P[h]$ is Gaussian the average over $h(x)$ can be performed explicitly and we get

$$W_n[K_i] := \int \mathcal{D}h \, P[h] Z^n[h, K^i] = \int \prod_{a=1}^n \mathcal{D}\mu_a \, e^{-S_{\mathrm{rep}} + \sum_{i,a} \int K_i \mathcal{O}_{i,a}}, \tag{76}$$

where

$$S_{\mathrm{rep}} = \sum_a S_{0,a} - \frac{v}{2} \sum_{a,b} \int d^d x \, \mathcal{O}_{0,a}(x)\mathcal{O}_{0,b}(x), \tag{77}$$

is the replica action. We see how a coupling between the replica theories has been generated after the average. Renormalization will possibly induce other couplings in the replica theory, all compatible with the symmetries of the system. Among these, importantly the replica theory enjoys an $S_n$ replica symmetry that permutes the various copies of the pure theory. We now assume that $W_n$ can be analytically continued for arbitrary values of $n$ including the origin in the complex $n$-plane.[15] Using (74) we find

$$W_D = \lim_{n \to 0} \left( \frac{\partial W_n}{\partial n} \right), \tag{78}$$

where $W_D$ is defined in (23), and thus

$$\overline{\langle \mathcal{O}_1(x_1)\mathcal{O}_2(x_2)\dots \rangle}_c = \lim_{n \to 0} \partial_n \left( \left\langle \sum_a \mathcal{O}_{1,a}(x_1) \sum_b \mathcal{O}_{2,b}(x_2)\dots \right\rangle^{\mathrm{rep}} \right), \tag{79}$$

---

[14]Normalization factors of $P[h]$, which ensure that probabilities add to one, will not play a role in our considerations and are then left implicit.

[15]This is a notoriously subtle limit. In particular we can have the phenomenon of spontaneous replica symmetry breaking (see [74] and references therein). We assume in what follows that the replica symmetry is not spontaneously broken.

where we used the fact that

$$\lim_{n\to 0} W_n[K_i] = 1. \tag{80}$$

Note that the in the left hand side of (79) we have the connected part of the correlator (indicated with the subscript $c$) which is computed in the replica theory by a suitable limit of a full correlator. Moreover, we see from (79) that a local operator $\mathcal{O}$ inside connected correlators of the disordered theory is mapped in the replica theory to its $S_n$-singlet component $\sum_a \mathcal{O}_a$.

The replica trick is also useful to compute general correlation functions in the disordered theory. Denoting by

$$S_a[h] = S_{0,a} + \int h(x)\mathcal{O}_{0,a}(x), \tag{81}$$

we have

$$
\begin{aligned}
\overline{\langle \mathcal{O}_1(x_1)\ldots\mathcal{O}_k(x_k)\rangle} &= \int \mathcal{D}h\, P[h] \frac{\int \mathcal{D}\mu\, e^{-S[h]}\mathcal{O}_1(x_1)\ldots\mathcal{O}_k(x_k)}{Z[h]} \\
&= \int \mathcal{D}h\, P[h] \frac{\int \prod_a \mathcal{D}\mu_a\, e^{-\sum_a S_a[h]}\mathcal{O}_{1,1}(x_1)\ldots\mathcal{O}_{k,1}(x_k)}{Z[h]^n},
\end{aligned}
\tag{82}
$$

which is an identity for any positive integer $n$. Assuming again that it can be analytically continued for $n \to 0$ we get[16]

$$\overline{\langle \mathcal{O}_1(x_1)\ldots\mathcal{O}_k(x_k)\rangle} = \lim_{n\to 0}\langle \mathcal{O}_{1,1}(x_1)\ldots\mathcal{O}_{k,1}(x_k)\rangle^{\text{rep}}. \tag{83}$$

In general correlators, in contrast to connected correlators, local operators are mapped to a specific copy (the same for all operators in the correlation function) in the replica theory. Equation (83) can easily be generalized to averages of products of general correlation functions. For example, omitting for simplicity the $x$-dependence of the local operators, we have

$$
\begin{aligned}
\overline{\left\langle \prod_{i=1}^{k}\mathcal{O}_i^{(1)}\right\rangle\left\langle \prod_{j=1}^{l}\mathcal{O}_j^{(2)}\right\rangle} &= \lim_{n\to 0}\int \mathcal{D}h\, P[h]\frac{\int \prod_a \mathcal{D}\mu_a\, e^{-\sum_a S_a[h]}\prod_{i=1}^{k}\mathcal{O}_{i,1}^{(1)}\prod_{j=1}^{l}\mathcal{O}_{j,2}^{(2)}}{Z^n[h]} \\
&= \lim_{n\to 0}\left\langle \prod_{i=1}^{k}\mathcal{O}_{i,1}^{(1)}\prod_{j=1}^{l}\mathcal{O}_{j,2}^{(2)}\right\rangle^{\text{rep}},
\end{aligned}
\tag{84}
$$

and similarly for more than two products. The last observables which we need to evaluate are averages of products of $N$ connected correlators. Before averaging, these correlators are obtained by taking functional derivatives of the product $W[K_i^{(1)}]\cdots W[K_i^{(N)}]$. For each of them we can use the replica trick to express this product as a unique path integral. We then have

$$\overline{\prod_{l=1}^{N}\left\langle \prod_{j_l=1}^{k_l}\mathcal{O}_{j_l}^{(l)}\right\rangle_c} = \left(\prod_{k=1}^{N}\lim_{n_k\to 0}\frac{\partial}{\partial n_k}\right)\left\langle \prod_{l=1}^{N}\prod_{j_l=1}^{k_l}\sum_{a_{j_l}^{(l)}=1}^{n_l}\mathcal{O}_{j_l,a_{j_l}^{(l)}}^{(l)}\right\rangle^{\text{rep}}, \tag{85}$$

where $S_{\text{rep}}$ is the replica theory for $n = \sum_{i=1}^{N} n_i$ replicas. Note that averages of products of general or connected correlators in the disordered theory can always be expressed in the replica theory as suitable limits of a *single general* correlator. Since any correlator can be expanded in its connected components, (85) is actually sufficient to compute generic correlation functions of the disordered theory. Any operator of the disordered theory gives rise to a multiplet transforming in the $n$-dimensional (natural) representation of $S_n$. Averages of connected correlators

---

[16]Note that we have actually taken the limit $n \to 0$ in the denominator of (82) ($Z^n[h] \to 1$) before integrating over $h$, while in the numerator it is kept after the integration over $h$.

of operators of the disordered theory are given by the $S_n$ singlet operators inside the natural representation in the replica theory. More general correlation functions of the disordered theory are instead given by considering operators singlets under subgroups $S_{n_i} \subset S_n$ induced by the natural representation in the replica theory.

## 3.2 Disordered symmetries from replica theory

Our first task is to understand how disordered symmetries manifest themselves in the replica theory. For concreteness we consider again the case of a $G = U(1)$ symmetry, the replica action reads

$$S_{\text{rep}} = \sum_a S_{0,a} - \frac{\nu}{2} \sum_{a,b} \int d^d x \, \mathcal{O}_{0,a}(x) \overline{\mathcal{O}}_{0,b}(x). \tag{86}$$

The $U(1)^n$ symmetry of the replicated pure part is broken by the disorder coupling to its diagonal $U(1)$ subgroup, which is then a symmetry of the replica theory. In particular there is a conserved current

$$J_D^\mu = \sum_a J_a^\mu, \tag{87}$$

constructed as the $S_n$ singlet out of the multiplet induced by the current $J^\mu$ of the disordered theory.

We can recover the Ward identities of the disordered symmetry from those produced by $J_D^\mu$ in the replica theory by using (85) for averages of products of connected correlators. The general key idea is to write a sum of averages of products of connected correlators with current insertions that, once mapped to correlators of the replica theory, reconstruct the complete diagonal current $J_D^\mu$. Then we can use the Ward identity in the replica theory and finally we rewrite the results back in terms of the disordered theory.

Determining the Ward identities for averages of single connected correlators is simple, because the diagonal current $J_D$ appears directly in the replica theory and we can immediately use the ordinary Ward identities there. We have

$$\overline{\left\langle \partial_\mu J^\mu(x) \mathcal{O}_1(x_1) \mathcal{O}_2(x_2) \cdots \right\rangle_c} = \lim_{n \to 0} \partial_n \left( \left\langle J_D^\mu(x) \sum_a \mathcal{O}_{1,a}(x_1) \sum_b \mathcal{O}_{2,b}(x_2) \dots \right\rangle^{\text{rep}} \right)$$

$$= \sum_i q_i \delta^{(d)}(x - x_i) \lim_{n \to 0} \partial_n \left\langle \sum_a \mathcal{O}_{1,a}(x_1) \sum_b \mathcal{O}_{2,b}(x_2) \dots \right\rangle^{\text{rep}}$$

$$= \sum_i q_i \delta^{(d)}(x - x_i) \overline{\left\langle \mathcal{O}_1(x_1) \mathcal{O}_2(x_2) \dots \right\rangle_c}, \tag{88}$$

which reproduces the connected version of (41). Averages of products of connected correlators are also easy to treat, because it is enough to consider a sum of correlators where the current is inserted in each term to reconstruct $J_D$ in the replica theory and then use the Ward identities there. Skipping obvious steps, we get

$$\sum_{j=1}^N \overline{\left\langle \partial_\mu J^\mu(x) \mathcal{O}_1^{(j)} \dots \mathcal{O}_{k_j}^{(j)} \right\rangle_c \left( \prod_{l \neq j} \left\langle \mathcal{O}_1^{(l)} \dots \mathcal{O}_{k_l}^{(l)} \right\rangle_c \right)} = \sum_{j=1}^N \sum_{i_j=1}^{k_j} \delta_{i_j^{(j)}} q_{i_j}^{(j)} \overline{\prod_{l=1}^N \left\langle \mathcal{O}_1^{(l)} \dots \mathcal{O}_{k_l}^{(l)} \right\rangle_c}, \tag{89}$$

which is similar to (45), but expressed in terms of connected correlators and the unshifted current.

Due to the different way the replica trick handles connected and general correlators, determining the Ward identities for the latter will produce the improved current $\widetilde{J}_\mu$. We use (83)

to write

$$
\begin{aligned}
\overline{\langle \partial^\mu J_\mu \mathcal{O}_1 \cdots \mathcal{O}_n \rangle} &= \lim_{n\to 0} \langle \partial^\mu J_{\mu,1} \mathcal{O}_{1,1} \ldots \mathcal{O}_{k,1} \rangle^{\mathrm{rep}} \\
&= \lim_{n\to 0} \langle \partial^\mu J_{\mu,1} \mathcal{O}_{1,1} \ldots \mathcal{O}_{k,1} \rangle^{\mathrm{rep}} - \lim_{n\to 0} \frac{1}{n-1} \left\langle \sum_{a=2}^n \partial^\mu J_{\mu,a} \mathcal{O}_{1,1} \ldots \mathcal{O}_{k,1} \right\rangle^{\mathrm{rep}} \quad (90) \\
&\quad + \lim_{n\to 0} \langle \partial^\mu J_{\mu,2} \mathcal{O}_{1,1} \ldots \mathcal{O}_{k,1} \rangle^{\mathrm{rep}}.
\end{aligned}
$$

In the last step, the last two terms add to zero due to the $S_n$ symmetry enjoyed by the replica theory. In the limit $n \to 0$ we have

$$
\lim_{n\to 0} \langle \partial^\mu J_{\mu,1} \mathcal{O}_{1,1} \ldots \mathcal{O}_{k,1} \rangle^{\mathrm{rep}} - \lim_{n\to 0} \frac{1}{n-1} \left\langle \sum_{a=2}^n \partial^\mu J_{\mu,a} \mathcal{O}_{1,1} \ldots \mathcal{O}_{k,1} \right\rangle^{\mathrm{rep}} = \lim_{n\to 0} \langle \partial^\mu J_{\mu,D} \mathcal{O}_{1,1} \ldots \mathcal{O}_{k,1} \rangle^{\mathrm{rep}},
$$

(91)

and

$$
\lim_{n\to 0} \langle \partial^\mu J_{\mu,2} \mathcal{O}_{1,1} \ldots \mathcal{O}_{k,1} \rangle^{\mathrm{rep}} = \overline{\langle \partial^\mu J_\mu \rangle} \, \overline{\langle \mathcal{O}_1 \cdots \mathcal{O}_k \rangle}. \tag{92}
$$

Therefore, by using the standard Ward identities of the replica theory, from (90) we get (41), as expected. The Ward identities (45) for products of generic correlators can be derived using a similar treatment:

$$
\begin{aligned}
\sum_{j=1}^N \overline{\left\langle \partial_\mu J^\mu \mathcal{O}_1^{(j)} \cdots \mathcal{O}_{k_j}^{(j)} \right\rangle} \left( \prod_{l\neq j} \overline{\left\langle \mathcal{O}_1^{(l)} \cdots \mathcal{O}_{k_l}^{(l)} \right\rangle} \right) &= \lim_{n\to 0} \sum_{j=1}^N \left\langle \partial^\mu J_{\mu,j} \prod_{j=1}^N \left( \mathcal{O}_{1,j}^{(j)} \cdots \mathcal{O}_{k_j,j}^{(j)} \right) \right\rangle^{\mathrm{rep}} \\
&= \lim_{n\to 0} \sum_{j=1}^N \left\langle \partial^\mu J_{\mu,j} \prod_{j=1}^N \left( \mathcal{O}_{1,j}^{(j)} \cdots \mathcal{O}_{k_j,j}^{(j)} \right) \right\rangle^{\mathrm{rep}} - \lim_{n\to 0} \frac{N}{n-N} \left\langle \sum_{a=N+1}^n \partial^\mu J_{\mu,a} \prod_{j=1}^N \left( \mathcal{O}_{1,j}^{(j)} \cdots \mathcal{O}_{k_j,j}^{(j)} \right) \right\rangle^{\mathrm{rep}} \\
&\quad + \lim_{n\to 0} N \left\langle \partial^\mu J_{\mu,N+1} \prod_{j=1}^N \left( \mathcal{O}_{1,j}^{(j)} \cdots \mathcal{O}_{k_j,j}^{(j)} \right) \right\rangle^{\mathrm{rep}} \quad (93) \\
&= \sum_{j=1}^N \sum_{i_j=1}^{k_j} q_{i_j}^{(j)} \delta^{(d)}(x - x_{i_j}^{(j)}) \overline{\prod_{l=1}^N \left\langle \mathcal{O}_1^{(l)} \cdots \mathcal{O}_{k_l}^{(l)} \right\rangle} + N \overline{\langle \partial^\mu J_\mu \rangle} \prod_i \overline{\left\langle \mathcal{O}_1^{(i)} \cdots \mathcal{O}_{k_i}^{(i)} \right\rangle}.
\end{aligned}
$$

The last term in the right-hand-side in the third row of (93) precisely combines with the left-hand-side to reproduce the shifted current $\widetilde{J}_\mu$ and hence the Ward identities (45).

The above analysis shows that the replica counterpart of the disordered symmetry is an ordinary symmetry generated by the diagonal current $J_D^\mu$ and all the Ward identities of the disordered theory reduce to Ward identities involving $J_D^\mu$ in the replica theory. The exotic selection rules (see discussion around (46)) of the disordered symmetry are a consequence of the non-trivial map between the observables of the replica theory and those in the theory with quenched disorder.

## 4 Disordered emergent symmetries and LogCFTs

Our analysis of Ward identities in section 2.2 applies for disordered symmetries, namely symmetries which are present in the underlying UV theory, are broken by the disorder, and get restored after disorder average. On the other hand, as in pure theories, we can have genuinely emergent symmetries in the IR, namely symmetries which are not present in the UV theory even before adding the disorder coupling. If the symmetry emerges for each theory in the ensemble, then we expect that it gives rise to approximate selection rules of the same

kind as in pure theories with emergent symmetries in the IR. However, we could also have symmetries that emerge in the IR only *after* disorder average. By definition, this implies the existence of additional selection rules which are valid on average in the IR of the theory. For non-emergent, actual disordered symmetries such selection rules arise from a conserved current which is a shifted version of the current operator $J^\mu$ of the UV theory $\widetilde{J}^\mu = J^\mu - \langle J^\mu \rangle$. For emergent symmetries we cannot determine its explicit form, as the description in terms of the UV action is useless, and the analysis in section 2.2 does not hold. However, as we will see, we can deduce which are the selection rules that the emergent disordered symmetry imposes on averaged correlation functions using the replica theory.

From a symmetry point of view, the key qualitative feature of the replica theory (for any finite $n$) is the presence of a $S_n$ global permutation symmetry not present in the original theory with disorder. In the analysis in section 3.2 the internal symmetry $G$ generated by the current $J_D^\mu$ commutes with $S_n$, namely the infinitesimal transformations $\delta\mathcal{O}_{j,a}$ of the fields do not mix different replicas. This is guaranteed by the fact that $G$ in the replica theory is the diagonal subgroup of the $G^n$ global symmetry of the replica theories when $v = 0$. On the other hand, in the case of an emergent symmetry this is not necessarily the case: each irreducible representation of $S_n$ can sit in a different $G$-representation, or even more generally, the local operators could sit in representations of the semi-direct product $G \rtimes S_n$. We expect that emergent symmetries in the replica theory of this kind correspond to *disordered emergent symmetries* in the theory with disorder. As we will see below, even in the deep IR the resulting selection rules will be modified with respect to those coming from (41) and its generalizations. As an application we will show how these modified Ward Identities allow for logarithmic conformal field theories (LogCFTs) as IR fixed points of disordered systems.

## 4.1 Emergent disordered symmetries

Let us analyze in some detail the Ward Identities for emergent symmetries in the replica theory. We study theories in which the total symmetry is a direct product $G \times S_n$, since this particular case already exhibits interesting features. For further simplification, we consider $G = U(1)$ and correlators where only the singlet and the standard representations of $S_n$ are involved. Generalizations to other representations of $S_n$ or more general groups $G$ should be straightforward.

Consider the average of a single correlation function of $k$ local operators in the disordered theory. We consider both the general and the connected part of the correlator. Using (83) and (79), they are mapped in the replica to the $n \to 0$ limit of respectively $\langle \mathcal{O}_{1,1} \ldots \mathcal{O}_{k,1} \rangle^{\text{rep}}$ and $\partial_n \langle \sum_{a_1} \mathcal{O}_{1,a_1} \ldots \sum_{a_k} \mathcal{O}_{k,a_k} \rangle^{\text{rep}}$, omitting the space dependence of the operators in the correlators for simplicity. The replica theory is an ordinary pure theory and the emergent symmetry should manifest with the existence of a vector local operator $J_D^\mu$, which becomes conserved in the IR. The operator $J_D^\mu$ is necessarily a singlet of $S_n$, since $U(1)$ commutes with $S_n$ by definition. Note that we do not need to assume the knowledge of the full multiplet $J_a^\mu$ for which $J_D^\mu = \sum_{a=1}^n J_a^\mu$. Indeed, while in the UV, for weak disorder, the existence of vector operators in the natural representation of $S_n$ is guaranteed, we do not need to keep track of the IR fate of the non-singlet components. Assuming that $J_D^\mu$ is conserved in the IR also at *finite $n$*, the

following standard selection rules on $k$-point correlators apply:[17]

$$\sum_{j=1}^{k}\left\langle \sum_{a_j=1}^{n}\delta\mathcal{O}_{j,a_j}\prod_{j\neq i=1}^{k}\sum_{a_i=1}^{n}\mathcal{O}_{i,a_i}\right\rangle^{\text{rep}}=0\,,\tag{94}$$

$$\sum_{j=1}^{k}\left\langle \delta\mathcal{O}_{j,1}\prod_{j\neq i=1}^{k}\mathcal{O}_{i,1}\right\rangle^{\text{rep}}=0\,.\tag{95}$$

The key point is now to look more closely to the variations $\delta\mathcal{O}_{j,a_j}$. Indeed, the natural representation of $S_n$ is reducible and the $\mathcal{O}_i$'s split in

$$\mathcal{O}_i^{(S)}=\sum_{a=1}^{n}\mathcal{O}_{i,a}\,,\qquad \mathcal{O}_{i,a}^{(F)}=\mathcal{O}_{i,a}-\frac{1}{n}\mathcal{O}_i^{(S)}\,,\tag{96}$$

which transform in the singlet and in the standard, or fundamental, representation respectively.[18] The $U(1)$ symmetry acts as

$$\delta\mathcal{O}_i^{(S)}=q_{S,i}\mathcal{O}_i^{(S)}\,,\qquad \delta\mathcal{O}_{i,a}^{(F)}=q_{F,i}\mathcal{O}_{i,a}^{(F)}\,,\tag{97}$$

where the charges are generically different, $q_{S,i}\neq q_{F,i}$, and can possibly depend on $n$. The variations entering the Ward identities of the replica theory are then

$$\delta\mathcal{O}_{i,a}=\delta\mathcal{O}_{i,a}^{(F)}+\frac{1}{n}\delta\mathcal{O}_i^{(S)}=q_{F,i}\mathcal{O}_{i,a}+\frac{\Delta q_i}{n}\sum_{a=1}^{n}\mathcal{O}_{i,a}\,,\tag{98}$$

where

$$\Delta q_i:=q_{S,i}-q_{F,i}\,.\tag{99}$$

Since in connected correlators we only have singlet components, plugging (98) in (94) gives simply

$$\sum_{j=1}^{k}q_{S,j}\left\langle\prod_{i=1}^{k}\sum_{a_i=1}^{n}\mathcal{O}_{i,a_i}\right\rangle^{\text{rep}}=0\,.\tag{100}$$

On the other hand, plugging (98) in (95) equals

$$\begin{aligned}
0&=\sum_{j=1}^{k}q_{F,j}\left\langle\prod_{i=1}^{k}\mathcal{O}_{i,1}\right\rangle^{\text{rep}}+\sum_{j=1}^{k}\frac{\Delta q_j}{n}\left\langle\sum_{b=1}^{n}\mathcal{O}_{j,b}\prod_{j\neq i=1}^{k}\mathcal{O}_{i,1}\right\rangle^{\text{rep}}\\
&=\sum_{j=1}^{k}\left(q_{F,j}+\frac{\Delta q_j}{n}\right)\left\langle\prod_{i=1}^{k}\mathcal{O}_{i,1}\right\rangle^{\text{rep}}+\sum_{j=1}^{k}\frac{\Delta q_j}{n}\left\langle\sum_{b=2}^{n}\mathcal{O}_{j,b}\prod_{j\neq i=1}^{k}\mathcal{O}_{i,1}\right\rangle^{\text{rep}}\\
&=\sum_{j=1}^{k}q_{F,j}\left\langle\prod_{i=1}^{k}\mathcal{O}_{i,1}\right\rangle^{\text{rep}}+\sum_{j=1}^{k}\Delta q_j\left\langle\mathcal{O}_{j,2}\prod_{j\neq i=1}^{k}\mathcal{O}_{i,1}\right\rangle^{\text{rep}}\\
&\quad+\frac{1}{n}\sum_{j=1}^{k}\Delta q_j\left(\left\langle\prod_{j\neq i=1}^{k}\mathcal{O}_{i,1}\right\rangle^{\text{rep}}-\left\langle\mathcal{O}_{j,2}\prod_{j\neq i=1}^{k}\mathcal{O}_{i,1}\right\rangle^{\text{rep}}\right)\,.
\end{aligned}\tag{101}$$

---

[17]This is not a trivial assumption. In particular it is not satisfied by a scalar free theory with random field disorder, which has a discontinuity at $n=0$ and it flows to a CFT only for $n=0$ [22, 23].

[18]More general representations arise for composite operators of the disordered theory which, once replicated, correspond to multiplets of $S_n$ transforming in a (reducible) tensor product of two or more natural representations.

The existence of the limit $n \to 0$ requires that

$$\Delta q_j(n) = nK_j + O\left(n^2\right), \qquad \text{as} \quad n \to 0, \tag{102}$$

where

$$K_j = \left.\frac{\partial \Delta q_j}{\partial n}\right|_{n=0}. \tag{103}$$

We can use (102) to go back to the averaged correlators of the disordered theory and obtain the desired selection rules

$$\sum_{j=1}^{k} q_j \overline{\left\langle \prod_{i=1}^{k} \mathcal{O}_i \right\rangle} + \sum_{j=1}^{k} K_j \left( \overline{\left\langle \prod_{i=1}^{k} \mathcal{O}_i \right\rangle} - \overline{\langle \mathcal{O}_j \rangle \left\langle \prod_{j\neq i=1}^{k} \mathcal{O}_i \right\rangle} \right) = 0, \tag{104}$$

$$\sum_{j=1}^{k} q_j \overline{\left\langle \prod_{i=1}^{k} \mathcal{O}_i \right\rangle}_c = 0, \tag{105}$$

where

$$q_j = q_{F,j}|_{n=0} = q_{S,j}|_{n=0}, \quad j = 1,\dots,k. \tag{106}$$

A similar analysis can be repeated for averages of products of correlation functions of the kind (43). We report here only the final result:

$$\sum_{m=1}^{N} \sum_{j=1}^{k_m} \left[ \left(q_j^{(m)} + K_j^{(m)}\right) \overline{\prod_{l=1}^{N} \langle \Upsilon^{(l)} \rangle} \right.$$

$$\left. + K_j^{(m)} \left( \sum_{a\neq m} \overline{\langle \Upsilon_j^{(m)} \rangle \langle \mathcal{O}_j^{(m)} \Upsilon^{(a)} \rangle \prod_{l\neq m,a} \langle \Upsilon^{(l)} \rangle} - N \overline{\langle \mathcal{O}_j^{(m)} \rangle \langle \Upsilon_j^{(m)} \rangle \prod_{l\neq m} \langle \Upsilon^{(l)} \rangle} \right) \right] = 0, \tag{107}$$

where we introduced the notations

$$\Upsilon^{(l)} = \prod_{i=1}^{k_l} \mathcal{O}_i^{(l)}, \qquad \Upsilon_j^{(l)} = \prod_{i=1,i\neq j}^{k_l} \mathcal{O}_i^{(l)}, \tag{108}$$

with $K_j^{(m)}$ and $q_j^{(m)}$ the external sources and charges associated to $\mathcal{O}_j^{(m)}$. When $K_j = 0$, the selection rules (104) are the standard ones associated to a $U(1)$ conserved symmetry, while for $K_j \neq 0$ we get additional terms which affect the disconnected component of the correlator only, given that the connected part satisfies the ordinary selection rule (105). The fact that (105) holds implies that in the disordered theory we have a notion of operators $\mathcal{O}_i$ carrying a definite $U(1)$ charge $q_i$, yet in disconnected correlators some effect is responsible for the appearance of the extra terms proportional to $K_j$. It would be interesting to understand the origin of these extra factors directly from the disordered theory.

For $k = 2$, (104) and (105) simplify and can be rewritten as

$$(q_1 + q_2)\overline{\langle \mathcal{O}_1 \rangle_c \langle \mathcal{O}_2 \rangle_c} + (K_1 + K_2)\overline{\langle \mathcal{O}_1 \mathcal{O}_2 \rangle_c} = 0,$$
$$(q_1 + q_2)\overline{\langle \mathcal{O}_1 \mathcal{O}_2 \rangle_c} = 0. \tag{109}$$

If $K_1 + K_2 \neq 0$, independently of the value of $q_1 + q_2$, the connected part of the 2-point function has to vanish and only a disconnected component is allowed. We are not aware of disordered theories with $K_j \neq 0$ for an internal global symmetry. On the other hand, we will show in the next section that the exotic selection rules derived above, applied to the case of emergent conformal symmetry, are at the origin of the possible appearance of logarithmic CFTs in the IR of disordered theories.

## 4.2 LogCFTs

Theories with quenched disorder may flow to IR fixed points [60] described by LogCFTs. Such CFTs were first discussed in 2d [57,58], see e.g. [75] for a review or [59] for an introduction in general dimensions from an axiomatic point of view. It was recognized in [58] that LogCFTs are intrinsically associated in having primary operators that are highest weight of indecomposable but not irreducible representations of the conformal group. A derivation of how LogCFTs can arise as random fixed points was given in [60] and more recently in [66] by means of (suitable generalizations of) Callan-Symanzik equations, in both cases using replica methods. We provide here an alternative derivation, working out the generalization of (109) when the emergent group is assumed to be the conformal one.

In the IR fixed point of the replica theory we have a dilatation current $J_d^\mu$ which yields the topological dilatation operator

$$D\left[\Sigma^{(d-1)}\right] = \int_{\Sigma^{(d-1)}} J_d^\mu n_\mu \,. \tag{110}$$

The conformal Ward identities applied to a primary operator $\mathcal{O}$ imply

$$D\left[\Sigma_x^{(d-1)}\right]\mathcal{O}(x) = \delta_D\mathcal{O}(x) + \mathcal{O}(x)D\left[\Sigma_{\text{no }x}^{(d-1)}\right], \tag{111}$$

where

$$\delta_D\mathcal{O} = \left(\Delta + x^\mu\partial_\mu\right)\mathcal{O}(x), \tag{112}$$

$(\Sigma_{\text{no }x}^{(d-1)})$ $\Sigma_x^{(d-1)}$ is a closed codimension 1 surface (not) encircling $x$. The dilatation operator acts diagonally only on the irreducible representations (96):

$$\delta_D\mathcal{O}_i^{(S)}(x) = \left(\Delta_{S,i}(n) + x^\mu\partial_\mu\right)\mathcal{O}_i^{(S)}(x), \qquad \delta_D\mathcal{O}_{i,a}^{(F)}(x) = \left(\Delta_{F,i}(n) + x^\mu\partial_\mu\right)\mathcal{O}_{i,a}^{(F)}(x). \tag{113}$$

Thus on $\mathcal{O}_{i,a}(x)$ we have

$$\delta_D\mathcal{O}_{i,a}(x) = \left(\Delta_{F,i} + x^\mu\partial_\mu\right)\mathcal{O}_{i,a}(x) + \frac{\Delta_{S,i} - \Delta_{F,i}}{n}\sum_{\alpha=1}^n\mathcal{O}_{i,\alpha}(x), \tag{114}$$

where in general $\Delta_{S,i}(n) \neq \Delta_{F,i}(n)$ for finite $n$. We plug the above transformations in (95) with $k = 2$ and equal operators. In this way we find the analogues of (109) for scaling transformations:

$$\begin{aligned}
\left(x^\mu\partial_\mu + 2\Delta\right)\overline{\langle\mathcal{O}(x)\rangle_c\langle\mathcal{O}(0)\rangle_c} + 2K\overline{\langle\mathcal{O}(x)\mathcal{O}(0)\rangle_c} &= 0\,, \\
\left(x^\mu\partial_\mu + 2\Delta\right)\overline{\langle\mathcal{O}(x)\mathcal{O}(0)\rangle_c} &= 0\,,
\end{aligned} \tag{115}$$

where

$$\Delta := \Delta_F|_{n=0} = \Delta_S|_{n=0}, \qquad K = \partial_n(\Delta_S - \Delta_F)|_{n=0}\,. \tag{116}$$

The general solution of (115) reads

$$\begin{aligned}
\overline{\langle\mathcal{O}(x)\mathcal{O}(0)\rangle_c} &= \frac{c_1}{|x|^{2\Delta}}\,, \\
\overline{\langle\mathcal{O}(x)\mathcal{O}(0)\rangle} &= \frac{c_2}{|x|^{2\Delta}} - \frac{c_1\log(\mu|x|)}{|x|^{2\Delta}}\,,
\end{aligned} \tag{117}$$

where $c_{1,2}$ are two integration constants with mass dimension $-2\Delta$ and $\mu$ is an arbitrary mass scale. Note that in a LogCFT, due to the peculiar way dilatations act on operators, the presence of a mass scale is actually compatible with conformal symmetry (see e.g. [59] for a more

detailed explanation). We see that the log term arises when $K \neq 0$, which acts as a source term in the second equation in (115).

Whenever the LogCFT has some internal global symmetry $G$ which is not emergent in the IR but is an exact symmetry present along the whole RG flow (i.e. present for each member of the ensemble and not broken by the disorder), the derivation above shows that logarithms can only appear in two-point functions of operators singlets under $G$. Indeed, in the replica theory the symmetry $G$ gets replicated in $n$ (unbroken) copies $G_a$, while the conformal symmetry generally is not, being only emergent at the fixed point. A representation $\rho$ of $G$ acting on a primary operator $\mathcal{O}$ is then replicated into $n$ copies $\rho_a$, each acting only on $\mathcal{O}_a$. Let $g \in G_a$, by simple manipulations we get

$$
\begin{aligned}
\rho_a(g) \cdot \mathcal{O}^{(S)} &= \rho_a(g) \cdot \mathcal{O}_a - \mathcal{O}_a + \mathcal{O}^{(S)} \\
&= (\rho_a(g) - \mathbb{1}) \cdot \mathcal{O}_a^{(F)} + \frac{1}{n} (\rho_a(g) + (n-1)\mathbb{1}) \cdot \mathcal{O}^{(S)}.
\end{aligned}
\tag{118}
$$

Since $G_a$ are internal symmetries, which necessarily commute with the dilatation operator $D$, we have

$$
0 = [D, \rho_a(g)] \cdot \mathcal{O}^{(S)} = (\Delta_F - \Delta_S)(\rho_a(g) - \mathbb{1}) \cdot \mathcal{O}_a^{(F)}.
\tag{119}
$$

Unless $\rho$ is in the trivial representation, the only solution of (119) is

$$
\Delta_S(n) = \Delta_F(n),
\tag{120}
$$

which implies that the factor $K$ defined in (116) vanishes, and thus logharithms cannot appear in the two-point function of $\mathcal{O}$ at the IR fixed point.

## 5 Symmetries in ensemble average

We discuss in this section the case in which the random coupling is taken to be constant:

$$
h(x) \rightarrow h.
\tag{121}
$$

Such set-up, which does not physically describe impurities as in quenched disorder, is particularly interesting in the light of the recent understanding of the role of average QFTs in the AdS/CFT correspondence [34]. As in the case of quenched disorder, we are interested in the situation where a symmetry is explicitly broken in any element of the ensemble and we want to see when and under which conditions it can emerge after the average. To distinguish them from the case of disordered systems, we will call these symmetries *averaged symmetries*. A notable example of this kind is the $O(N)$ symmetry in the SYK model [31–33] which rotates the $N$ Majorana fermions, broken by the random fermion coupling, and restored after average (provided the average is taken with an $O(N)$-invariant distribution, as is often the case).

We will see that the simple replacement (121) leads to crucial differences with respect to the quenched disorder case. We discuss the importance of connectedness of the full space in section 5.1, we derive the Ward identities and the topological operators emerging after ensemble average in section 5.2, and finally in section 5.3 we comment on the implications of our results in the context of the AdS/CFT correspondence where the ensemble average is supposed to be the dual theory of a bulk theory of gravity in $d + 1$ dimensions.

### 5.1 Selection rules in disconnected spaces

The presence of a constant random coupling $h$ over the entire space $X^{(d)}$ leads to a new effect, not present in the quenched disorder, which is the lack of factorization of correlation functions

in *disconnected* spaces. For definiteness, consider a theory deformed by a random coupling $h$ in a space $X^{(d)}$ which is the union of two spaces $X^{(d)} = X_1^{(d)} \sqcup X_2^{(d)}$, with $X_1^{(d)} \cap X_2^{(d)} = \emptyset$. At this stage we are not specifying whether the coupling is a constant or not, we only assume that it breaks a global 0-form symmetry $G$ of the pure theory. For each element of the ensemble we can define a generating functional introducing sources $K_i$ for the local operators $\mathcal{O}_i$. Since the space manifold is disconnected, for each local operator $\mathcal{O}$ we effectively need two sources, $K_1$ and $K_2$, defined in $X_1^{(d)}$ and $X_2^{(d)}$. For any $h$, constant or not, the total functional factorizes[19]

$$Z[X^{(d)}, K, h] = Z[X_1^{(d)}, K_1, h]\, Z[X_2^{(d)}, K_2, h]\,, \tag{122}$$

and so will do arbitrary correlation functions of local operators $\Phi$:

$$\langle \Phi \rangle_X = \langle \Phi_1 \rangle_{X_1} \langle \Phi_2 \rangle_{X_2}\,, \tag{123}$$

where $\Phi_i$ is the subset of operators in $\Phi$ supported on $X_i$. When $h$ is space dependent (quenched disorder), its support and its probability measure splits into $X_1$ and $X_2$. Hence quenched averaged correlators factorize in the two distinct components:[20]

$$\overline{\langle \Phi_1 \rangle_{X_1} \langle \Phi_2 \rangle_{X_2}} = \overline{\langle \Phi_1 \rangle}_{X_1}\, \overline{\langle \Phi_2 \rangle}_{X_2}\,. \tag{124}$$

Thanks to this factorization, the selection rules of the disordered theory are realized independently on each connected component:

$$\overline{\langle \Phi_i \rangle}_{X_i} = R_i\, \overline{\langle \Phi_i \rangle}_{X_i}\,, \qquad i = 1, 2 \quad \text{(quenched disorder)}\,, \tag{125}$$

where $R_i$ are the direct products of the representations of the local operators in $X_i^{(d)}$, which should each contain a singlet to get a non-vanishing correlator.

Crucially, in the ensemble average case (124) cannot hold, because a constant $h$ does not split on the connected components and the average *correlates* the operators across $X_1^{(d)}$ and $X_2^{(d)}$. In particular, we now get the selection rules

$$\overline{\langle \Phi_1 \rangle_{X_1} \langle \Phi_2 \rangle}_{X_2} = R_1 \cdot R_2\, \overline{\langle \Phi_1 \rangle_{X_1} \langle \Phi_2 \rangle}_{X_2} \quad \text{(ensemble average)}\,. \tag{126}$$

In contrast to the quenched disorder case, *averages of single correlators in the ensemble average effectively turn into averages of products of correlators when the space is disconnected*. The constraint (126) is weaker than (125), obtained in the quenched average theory. In (126) we need the singlet to appear only in the product $R_1 \cdot R_2$, in (125) separately for $R_1$ and $R_2$. For symmetries that emerge after ensemble average, which we dub *average* symmetries, the charge is then *not* conserved on a single connected component of the manifold, but can "escape" to the other connected components (see the end of appendix A.1.2 for an explicit computation in a free scalar model). We will see how this relates to the violation of global symmetries by Euclidean wormholes in section 5.3. The above analysis is trivially generalized to a space with an arbitrary number of disconnected components and to arbitrary products of correlation functions of local operators.

---

[19]This follows from the observation that any map whose domain is disconnected can be written uniquely as a sum of maps each supported in a connected component.

[20]It should not be confused this factorization of correlators in disconnected space with the non-factorization of products of averaged correlators due to quenched disorder considered in section 2 and present in any space $X^{(d)}$, connected or not.

## 5.2 Ensemble average and Ward identities

The analysis presented in section 2.2 can be repeated in the case of constant $h$. For concreteness we consider again the case in which the pure theory has a $U(1)$ global symmetry under which $\mathcal{O}_0$ has charge $q_0$. We have one complex parameter $h$ and the average generating functional is

$$\overline{Z[K_i]} = \int \mathrm{d}h \mathrm{d}\bar{h} \, P[\bar{h}h] \frac{\int \mathcal{D}\mu e^{-S_0-(h\int \mathcal{O}_0+c.c.)+\int K_i \mathcal{O}_i}}{\int \mathcal{D}\mu e^{-S_0-(h\int \mathcal{O}_0+c.c.)}}\,. \tag{127}$$

We derive identities between correlators by changing variables inside the various integrals in (127). By changing variable in the numerator with an infinitesimal space-dependent symmetry transformation of parameter $\epsilon(x)$, we get

$$\langle \partial_\mu J^\mu(x)\Phi\rangle = \sum_i \delta^{(d)}(x-x_i)q_i\langle\Phi\rangle + q_0\langle \mathcal{D}(x;h)\Phi\rangle\,, \tag{128}$$

where the sum runs over all the local operators defining $\Phi$ and we have defined

$$\mathcal{D}(x;h) := -h\mathcal{O}_0(x) + \bar{h}\overline{\mathcal{O}}_0(x)\,. \tag{129}$$

Note that (128) holds *before* taking the average. Indeed, this is nothing else than the Ward identities one obtains in a pure theory for an explicitly broken symmetry. We are now not allowed to do a change of variable in the $h$ integral to possibly prove the vanishing on average of the last term in (128). However, we can perform a *global* transformation $h \to e^{-iq_0\epsilon}h$, with $\epsilon$ constant, inside (127). In this way, we get

$$\int_{X^{(d)}} \overline{\langle \mathcal{D}(x;h)\Phi\rangle} = \int_{X^{(d)}} \overline{\langle \mathcal{D}(x;h)\rangle\langle\Phi\rangle}\,, \tag{130}$$

where $X^{(d)}$ is the *full* space manifold. Finally we can perform a space dependent $U(1)$ transformation only in the path integral in the denominator of (127), getting

$$\langle \partial_\mu J^\mu\rangle = q_0\langle \mathcal{D}\rangle\,, \tag{131}$$

valid before ensemble average. From now on we will assume that $\mathcal{O}_0$ is a scalar under spatial rotations,[21] so that every element of the ensemble is $\mathfrak{so}(d)$ invariant. We then have $\langle J_\mu\rangle = 0$ and thanks to (131) the relation (130) simplifies to

$$\int_{X^{(d)}} \overline{\langle \mathcal{D}(x;h)\Phi\rangle} = 0\,. \tag{132}$$

See appendix A.1.2 for an explicit derivation of (132) for a two-point function in a simple solvable model. The combination $\partial^\mu J_\mu - q_0\mathcal{D}(x)$ satisfies the condition

$$\int_{X^{(d)}} d^d x \, \overline{\langle \left(\partial^\mu J_\mu(x) - q_0\mathcal{D}(x;h)\right)\Phi\rangle} = 0\,, \tag{133}$$

which ensures that the Ward identities (128), when integrated over the full space and after ensemble average, imply charge conservation. As expected from a spurionic argument, the symmetry is restored after average.[22]

---

[21]This assumption is not crucial. For non-scalar deformations, rotational invariance is broken before the average and we need to keep track of all the vacuum expectation values induced by the random variable, as done in the quenched disorder case. This can be repeated in the ensemble average case, but makes the analysis more involved.

[22]In a pure theory the identities (128) apply but $\mathcal{D}(x)$ does not integrate to zero when inserted in arbitrary correlators. As a consequence no selection rules are implied, as expected for an explicitly broken symmetry!

Let us now see if we can define more general operators $\widehat{Q}\left[\Sigma^{(d-1)}, D^{(d)}; h\right]$, topological after ensemble average. The natural choice from (133) is

$$
\widehat{Q}\left[\Sigma^{(d-1)}, D^{(d)}; h\right] = Q\left[\Sigma^{(d-1)}\right] - q_0 \int_{D^{(d)}} d^d x \, \mathcal{D}(x; h),
$$

$$
Q\left[\Sigma^{(d-1)}\right] := \int_{\Sigma^{(d-1)}} n_\mu J^\mu(x),
$$

(134)

where $D^{(d)}$ is an arbitrary region such that $\partial D^{(d)} = \Sigma^{(d-1)}$. Note that this requires $\Sigma^{(d-1)}$ to be homologically trivial otherwise, by definition, the surface $D^{(d)}$ does not exist. In the terminology of [76], the operator (134) is a *non-genuine* co-dimension one operator, since it requires a topological surface attached to it.[23]

We can discuss the dependence of $\widehat{Q}$ in (134) on the choice of the filling region $D^{(d)}$. Given another such manifold $D'^{(d)}$ we can glue it along $\Sigma^{(d-1)}$ with the orientation reversal of $D^{(d)}$ to form a closed manifold $Y^{(d)} = D'^{(d)} \sqcup \overline{D^{(d)}}$, and $\widehat{Q}[\Sigma^{(d-1)}, D^{(d)}]$ is independent on $D^{(d)}$ if and only if

$$
\int_{Y^{(d)}} \overline{\left\langle \mathcal{D}(x; h) \Phi \right\rangle} = 0.
$$

(135)

We see that (135) is not satisfied unless the space-time $X^{(d)}$ is connected, and we will generically refer to $\widehat{Q}$ as a non-genuine operator. On the other hand, if $X^{(d)}$ is connected any homologically trivial co-dimension one submanifold $\Sigma^{(d-1)}$ of $X^{(d)}$ divides $X^{(d)} - \Sigma^{(d-1)}$ in two disjoint connected components glued along $\Sigma^{(d-1)}$, hence necessarily $Y^{(d)} = X^{(d)}$ and (135) reduces to (132), showing the independence of $\widehat{Q}[\Sigma^{(d-1)}, D^{(d)}; h]$ on the filling region. $\widehat{Q}$ is still expressed with an integral over $D^{(d)}$, but the dependence of the non-genuine symmetry operator on the filling region is only apparent, and for all practical purposes this can be regarded as independent on the filling region. We refer to this situation as a quasi-genuine co-dimension one operator.

If $X^{(d)}$ has several connected components, $Y^{(d)}$ can be a proper sub-region, since adding or removing from it an entire connected component which does not intersect $\Sigma^{(d-1)}$ preserves the property that $Y^{(d)}$ is the union of regions glued along $\Sigma^{(d-1)}$. For instance if $X^{(d)}$ has two connected components $X_1^{(d)}$ and $X_2^{(d)}$, and suppose $\Sigma^{(d-1)}$ is entirely contained in $X_1^{(d)}$, the latter is divided by $\Sigma^{(d-1)}$ into two regions $D^{(d)}$ and $D'^{(d)}$, and choosing one or the other leads to different operators $\widehat{Q}[\Sigma^{(d-1)}; h]$, since (132) holds only in the entire space and not to each connected component:

$$
\overline{\left\langle \int_{D^{(d)}} \mathcal{D}(x; h) \Phi \right\rangle} = \overline{\left\langle \left( \int_{D'^{(d)}} + \int_{X_2^{(d)}} \right) \mathcal{D}(x; h) \Phi \right\rangle} \neq \overline{\left\langle \int_{D'^{(d)}} \mathcal{D}(x; h) \Phi \right\rangle}.
$$

(136)

In this case we cannot define a quasi-genuine co-dimension one topological operator and therefore, even if the total charge is conserved thanks to (133), we cannot measure it locally in a subregion of the entire (disconnected) space.

In order to measure the charge of operators in the whole space, we can consider $\widehat{Q}$ on a codimension 1 closed surface $\Sigma^{(d-1)} = \Sigma_1^{(d-1)} \sqcup \Sigma_2^{(d-1)}$, with $\Sigma_i^{(d-1)} \subset X_i^{(d)}$ $(i = 1, 2)$, and two regions $D_i^{(d)}$ such that $\partial D_i^{(d)} = \Sigma_i^{(d-1)}$. In each given connected component, the charge cannot be conserved, as we have seen, but if we simultaneously consider the two regions, then the

---

[23]The requirement is however of different nature. In [76] (and subsequent works) the surface is required to have a well-defined gauge-invariant operator, here the surface is required to make the operator topological (on average).

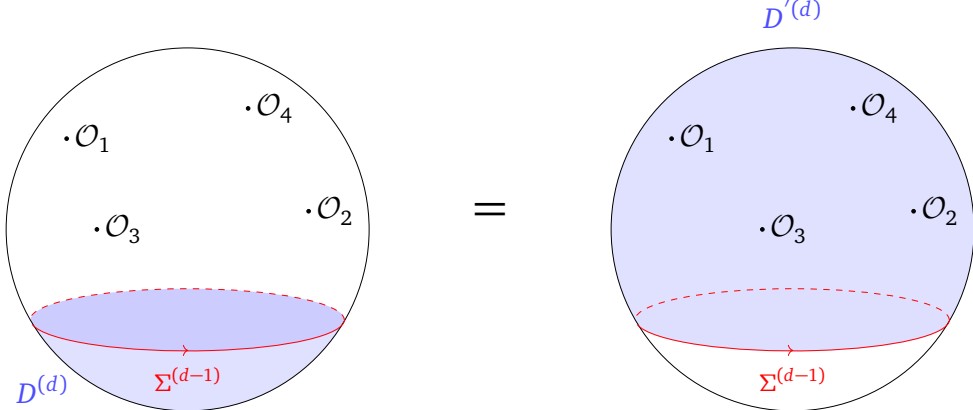

Figure 1: Selection rules (138) for correlators when $X^{(d)}$ is connected. The integral over the region $D^{(d)}$ in the left panel equals the integral over the region $D^{'(d)}$ in the right panel thanks to (132). When $D^{(d)}$ is shrunk to a point the region $D^{'(d)}$ extends to the whole $X^{(d)}$.

Ward identities still apply. In the schematic notation of section 5.1 we have

$$
\begin{aligned}
\overline{\langle \widehat{Q}[\Sigma, D; h]\Phi\rangle_X} &= \overline{\langle \widehat{Q}[\Sigma_1, D_1; h]\Phi_1\rangle_{X_1}\langle \widehat{\Phi}_2\rangle_{X_2}} + \overline{\langle \Phi_1\rangle_{X_1}\langle \widehat{Q}[\Sigma_2, D_2; h]\Phi_2\rangle_{X_2}} \\
&= \left(\chi_1(\Sigma_1) + \chi_2(\Sigma_2)\right)\overline{\langle \widehat{\Phi}_1\rangle_{X_1}\langle \widehat{\Phi}_2\rangle_{X_2}} = \left(\chi_1(\Sigma_1) + \chi_2(\Sigma_2)\right)\overline{\langle \widehat{\Phi}\rangle_X} ,
\end{aligned}
\tag{137}
$$

where $\chi_{1,2}(\Sigma_{1,2})$ denotes the sum of the charges of the local operators $\Phi_{1,2}$ which are inside the surface $\Sigma_{1,2}^{(d-1)}$. Since $\Sigma^{(d-1)}$ depends now on $D^{(d)}$, it is crucial to consider the complement space in both connected spaces at the same time. The generalization to spaces $X^{(d)}$ with more than two connected components is obvious.

We refer the reader to appendix B for a proof of the existence of the operator $\widehat{U}_g$ which implements the action of the group rather than the action of the corresponding Lie algebra. By definition, the operator $\widehat{U}_g$, given in (B.20), satisfies

$$
\overline{\langle \widehat{U}_g[\Sigma^{(d-1)}, D^{(d)}; h]\mathcal{O}_1 \cdots \mathcal{O}_n\rangle} = e^{i\alpha\chi(\Sigma^{(d-1)})}\overline{\langle \mathcal{O}_1 \cdots \mathcal{O}_n\rangle} .
\tag{138}
$$

Since $\widehat{U}_g[\emptyset, X^{(d)}; h] = 1$, (138) implies the selection rules we derived from the spurion argument (see figure 1). The equivalent of (137) for a finite group action precisely reproduces the selection rule (126). With $\Sigma^{(d-1)}$ as in figure 2, we have

$$
\begin{aligned}
\overline{\langle \Phi\rangle_X} &= \overline{\langle \widehat{U}_g[\Sigma, D; h]\Phi\rangle_X} = \overline{\langle \widehat{U}_g[\Sigma_1, D_1; h]\Phi_1\rangle_{X_1}\langle \widehat{U}_g[\Sigma_2, D_2; h]\Phi_2\rangle_{X_2}} \\
&= e^{i\alpha(\chi_1(\Sigma_1) + \chi_2(\Sigma_2))}\overline{\langle \Phi\rangle_X} ,
\end{aligned}
\tag{139}
$$

while, say,

$$
\overline{\langle \Phi\rangle_X} = \overline{\langle \widehat{U}_g[\Sigma_1, D_1; h]\Phi_1\rangle_{X_1}\langle \Phi_2\rangle_{X_2}} \neq e^{i\alpha\chi_1(\Sigma_1)}\overline{\langle \Phi\rangle_X} .
\tag{140}
$$

We have then found an instance of a theory with a global zero-form symmetry in the sense of giving rise to selection rules for correlation functions of local operators, but with *no* genuine codimension one topological operator. Aside of being topological only on average, the operator $\widehat{U}_g[\Sigma, D; h]$ is not genuine and it can be defined only on homologically trivial cycles.

The local charge violation (140) in a single connected component of space when $X^{(d)}$ is an union of several connected components indicate the presence of non-local interactions in the theory. Their presence is manifest by using the replica trick. Consider a Gaussian random

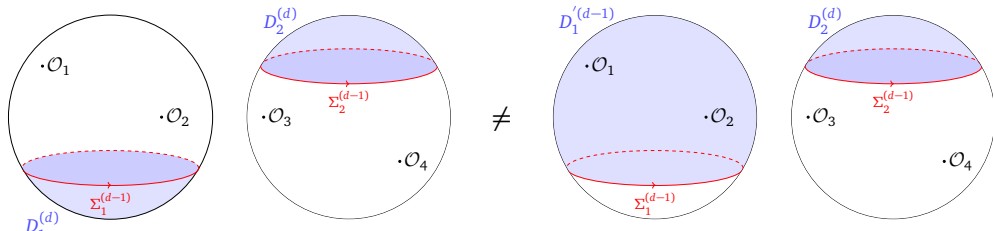

Figure 2: Violation of the selection rules (140) when $X^{(d)}$ is disconnected. The integral over the region $D_1^{(d)}$ in $X_1^{(d)}$ (left) is not equal to the integral over the region $D_1^{'(d)}$ in $X_1^{(d)}$ (right) because of the presence of the component $X_2^{(d)}$. An equality sign would require to reverse the region of integration also in $X_2^{(d)}$ (right) from $D_2^{(d)}$ to its complement.

distribution $P[\bar{h}h] \propto \exp(-\bar{h}h/v)$ (e.g. as in the SYK model). Repeating the steps described in section 3 we find *non-local* interactions among replicas

$$S_{\text{rep}} = \sum_{a=1}^{n} S_{0,a} - v \int d^d x \int d^d y \sum_{a,b=1}^{n} \overline{\mathcal{O}}_{0,a}(x)\mathcal{O}_{0,b}(y). \tag{141}$$

The replica theory enjoys a diagonal $U(1)_D$ global symmetry, but the naive diagonal current $J_D^\mu = \sum_a J_a^\mu$ does not satisfy standard Ward identities. By performing an infinitesimal $U(1)$ transformation with a local parameter $\alpha(x)$ we get

$$\delta S_{\text{rep}} = \int dx\, \alpha(x)\partial_\mu J_D^\mu(x) - q_0 v \sum_{a,b} \int dx\, dy \left(\alpha(y) - \alpha(x)\right)\overline{\mathcal{O}}_{0,a}(x)\mathcal{O}_{0,b}(y) \tag{142}$$

$$= \int_{X^{(d)}} dx\, \alpha(x)\left(\partial_\mu J_D^\mu(x) + q_0 v \sum_{a,b} \int_{X^{(d)}} dy \left(\overline{\mathcal{O}}_{0,a}(x)\mathcal{O}_{0,b}(y) - \mathcal{O}_{0,a}(x)\overline{\mathcal{O}}_{0,b}(y)\right)\right).$$

Thus the Ward identities for the diagonal symmetry are modified by a non-local term and read

$$\left\langle \left(\left(\partial_\mu J_D^\mu(x) + q_0 v \sum_{a,b} \int_{X^{(d)}} d^d y \left(\overline{\mathcal{O}}_{0,a}(x)\mathcal{O}_{0,b}(y) - \overline{\mathcal{O}}_{0,a}(y)\mathcal{O}_{0,b}(x)\right)\right)\Phi\right\rangle^{\text{rep}} = \sum_i \delta^{(d)}(x-x_i)q_i\langle\Phi\rangle^{\text{rep}}. \tag{143}$$

In the replica theory the operator

$$\partial_\mu J_D^\mu(x) + q_0 v \sum_{a,b} \int_{X^{(d)}} d^d y \left(\overline{\mathcal{O}}_{0,a}(x)\mathcal{O}_{0,b}(y) - \overline{\mathcal{O}}_{0,a}(y)\mathcal{O}_{0,b}(x)\right), \tag{144}$$

satisfies the Ward identities and its integral over the full space evidently vanishes (inside arbitrary correlators), implying the $U(1)_D$ selection rules. This is how the properties of the averaged symmetry show up in the replica theory, where the non-local nature of the symmetry is manifest for Gaussian distributions. The property (132) of the operator $\mathcal{D}(x)$ defined in (129) is mapped to the property of the extra term in (144) of integrating to zero exactly as an operator equation. This is consistent with the dictionary between correlators of the averaged theory and the replica one.

## 5.3 A gravity discussion

We have found that averaged global symmetries are intrinsically different from ordinary global symmetries. They imply selection rules as dictated by the global symmetry but, in contrast to

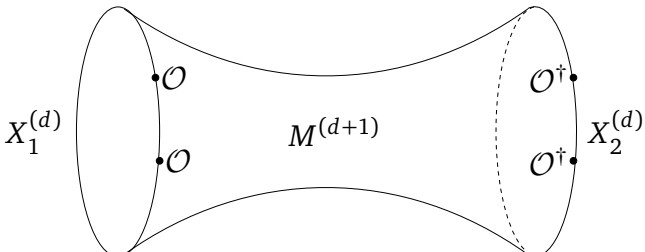

Figure 3: Example of a wormhole bulk geometry $M^{(d+1)}$ with boundary $X_1^{(d)} \sqcup X_2^{(d)}$ contributing to the average correlator $\overline{\langle \mathcal{O O} \rangle \langle \mathcal{O}^\dagger \mathcal{O}^\dagger \rangle}$, with $\mathcal{O}$ a charged boundary operator.

ordinary global symmetries, they do not admit genuine co-dimension one operators, topological after average. Even in a connected space such operators cannot be defined in homologically non-trivial cycles. As a result, these symmetries cannot consistently be coupled to an external background field, at least not in a natural way.[24] Note that this is different from the concept of 't Hooft anomalies. In the latter the obstruction is in making the gauge fields dynamical but there is a well defined notion of coupling the theory to backgrounds gauge fields. The difficulty of coupling the symmetry to an external background is clear in the replica theory from the presence of the second term in (144), which is non-local and not manifestly the divergence of a current.

The results have interesting consequences when applied to averaged theories which are assumed to have an holographic dual bulk gravitational theory in asymptotically AdS spacetimes.

In the ordinary AdS/CFT correspondence a given theory of gravity in asymptotically AdS space-time is dual to a given CFT. Ordinary global symmetries of the CFT become gauge symmetries in the bulk. This correspondence fits nicely with the widely accepted common lore that in quantum gravity unbroken global symmetries cannot exist [77–80]. A natural question then arises: when the dual theory is given by an ensemble average, what is the bulk interpretation of the symmetries emerging after average? In [61] (see also [62,63]) it has been conjectured that boundary emergent symmetries correspond in the bulk to global, and not gauge, symmetries which are broken non-perturbatively by Euclidean wormhole configurations, which allow the global symmetry charge to flow from one connected component to another one, see figure 3. From the boundary point of view, this charge violation induced by bulk wormholes correspond to the lack of selection rules in the average theory that we have discussed before, when the space is not connected, in agreement with the findings in [61–63]. Since averaged symmetries simply cannot be gauged, our results clarify why they cannot be interpreted as gauge symmetries in the bulk, at least in the case where the average is of the form (14).[25]

Note that boundary emergent symmetries are compatible with recent works where, motivated by the connection with the lore of spectrum completeness in gravitational theories [81], "absence of global symmetries in gravitational theories" is replaced by "absence of topological operators", including those related to non-invertible symmetries [82,83].

---

[24]For discrete symmetries, for example, coupling to an external background field corresponds to insert a mesh of symmetry defects on homologically non-trivial cycles of space.

[25]In particular, our results do not straightforwardly apply when the average is over OPE coefficients, as e.g. discussed in [44,45].

# 6 Conclusions

In this paper we have studied disordered QFTs where an ordinary symmetry of a pure QFT is explicitly broken by a random coupling, but the symmetry re-emerges after quenched average. We focused our attention to understand if and under what conditions we can have operators, topological on average, in analogy to ordinary QFTs [1]. We considered *quenched disorder theories*, where the pure theory is deformed with a space dependent coupling, and *ensemble average theories*, where the latter is kept constant.

In the quenched disordered case, we can write Ward identities for averages of products of correlators and construct the symmetry operator implementing the finite group action, topological after average. Such disordered symmetries can be coupled to external background, can be gauged, and can have 't Hooft anomalies (i.e. can exclude a trivially gapped phase at long distances), precisely like ordinary symmetries. Using the replica trick, we also discussed genuinely emergent symmetries in the IR after average, namely symmetries which are not present in the UV theory even before adding the disorder coupling. We pointed out that whenever a symmetry $G$ is emergent in the IR, exotic selection rules can explain the origin of LogCFTs.

In ensemble average theories the analogy to pure QFTs is more loose. We still have selection rules for averages of correlators and we can construct operators implementing the finite group action, but the charge operator is not purely codimension-1 and cannot be defined if $\Sigma^{(d-1)}$ is homologically non-trivial. When the space is disconnected, the selection rules apply only globally and in each connected component charge violation can occur. Such averaged symmetries cannot be coupled to background gauge fields in ordinary ways. The difficulty (impossibility) of gauging emergent boundary symmetries clarify why such symmetries cannot be identified with bulk gauge symmetries when the average theory admits a gravitational bulk dual.

It would be interesting to analyze spontaneous breaking of disordered symmetries in more detail.[26] There are essentially two ways in which the disordered symmetry could spontaneously break: i) the symmetry is spontaneously broken in the pure theory before adding the random interaction, ii) the symmetry is unbroken in the pure theory and the random interaction induces a spontaneous breaking of the disordered symmetry. Let us consider the case of continuous symmetries. From the replica theory point of view, i) and ii) are distinguished by which components of the replica currents $J_a^\mu$ are subject to spontaneous breaking, all components in case i) and only the singlet $\sum_a J_a^\mu$ in case ii). Assuming the existence of the analytic continuation in $n$ and of a smooth $n \to 0$ limit, we expect for $d > 2$ gapless excitations (Goldstone boosons) in the replica theory, giving rise to power-like correlators. From the disordered theory point of view, in case i) there is a Goldstone mode in the pure theory which acquires a mass in each specific realization of the ensemble, turning into a pseudo Goldstone boson. In contrast, no Goldstone boson is present in the pure theory in the more exotic case ii). In both cases it would be nice to identify which correlators (if any) exhibit power like-behavior on average as a result of the spontaneous breaking of the disordered theory.

It would be also interesting to generalize our findings to quantum disorder, namely to Lorentzian theories where the random coupling depends only on space. The natural extension of our analysis beyond 0-form symmetries does not seem straightforward. Higher-form symmetries can be broken only by non-local deformations, which should be also taken random. It is possibly easier to consider a set-up in $d = 2$ where non-invertible symmetries can be obtained by 0-form symmetries only, and see if and in what sense we can have a non-invertible symmetry re-emerging after average.

An important remark about the ensemble average case is that, in comparing our findings with the existing literature on the factorization problem in AdS/CFT, one should keep in mind

---

[26]We thank O. Aharony for a question that prompted the paragraph that follows.

that we only considered averaging over couplings. There are other setups, like averaging over OPE coefficients [44, 45] or over different modular invariants [84], where global symmetries could behave differently from our findings. In particular [84] discusses the gauging of a 1-form global symmetry in certain 3$d$ gravitational toy models, which are outside our set-up. Indeed the putative boundary dual of these toy models cannot be described as a random deformation of a pure theory, rather the partition function is an average of characters of a current algebra [84] which, taken singularly, do not define a physical theory. It is a very interesting problem for the future to discuss the status of global symmetries in these other contexts, possibly finding a unified picture.

# Acknowledgments

We thank G. Delfino, L. Di Pietro, Z. Ji, S. Rychkov, E. Trevisani for useful discussions. We also thank O. Aharony, T. Hartman and Z. Komargodski for comments on the manuscript. MS thanks the organizers of the Bootstrap 2022 conference held in Porto, where the idea of this project took form, and the Institut des Hautes Études Scientifiques (IHES), where part of this work has been done, for the hospitality.

**Funding information**   Work partially supported by INFN Iniziativa Specifica ST&FI. A.A. and G.R. are supported in part by the ERC-COG grant NP-QFT No. 864583 "Non-perturbative dynamics of quantum fields: from new deconfined phases of matter to quantum black holes".

# A   Toy model examples

In this appendix we test some formulas of the main text in simple solvable examples. We first discuss linear random couplings in free scalar theories and establish the validity of the generalized Ward identity (41) for 2-point functions both for the case of $h(x)$ (quenched disorder) and constant $h$ (ensemble average). Subsequently we test the 't Hooft anomaly matching condition discussed in section 2.4 by working out a specific example.

## A.1   Free scalar theories

We consider the toy example of a complex free scalar perturbed by a linear random coupling. The action is

$$S = \int d^d x \left( |\partial \phi|^2 + m^2 |\phi|^2 + h\phi(x) + \bar{h}\bar{\phi}(x) \right). \tag{A.1}$$

The coupling to $h$ explicitly breaks the $U(1)$ symmetry rotating $\phi$. Here $h$ can have or not a space dependence. In both cases we can write

$$Z[K, \bar{K}, h] = \exp \left( \int d^d x d^d y (\bar{h} + \bar{K}(x)) G(x - y)(h + K(y)) \right), \tag{A.2}$$

where $G(x-y)$ is the massive scalar propagator in flat space and $K, \bar{K}$ are the external sources for $\phi$ and $\bar{\phi}$, respectively. We consider a Gaussian distribution with variance $v$ and zero mean in order to simplify the expressions. In what follows we shall be sloppy with normalizations and overall constants which do not affect the main points we want to show.

### A.1.1 Quenched disorder

It is convenient to introduce a compact notation

$$(hG)_x := \int d^d w\, h(w) G(w-x), \qquad (G\bar{h})_y := \int d^d w\, G(y-w)\bar{h}(w),$$
$$G_{xy} := G(x-y), \qquad (GG)_{xy} := \int d^d w\, G(x-w) G(w-y), \tag{A.3}$$

so that, from (A.2), we get the one-point function

$$\langle \phi(x) \rangle = Z^{-1} \frac{\delta Z}{\delta K(x)} \Big|_{K=0} = (G\bar{h})_x. \tag{A.4}$$

Since translation invariance is broken, this is not a constant. Similarly, for two point functions,

$$\langle \phi(x)\phi(y) \rangle = Z^{-1} \frac{\delta^2 Z}{\delta K(x)\delta K(y)} \Big|_{K=0} = (G\bar{h})_x (G\bar{h})_y,$$
$$\langle \bar{\phi}(x)\phi(y) \rangle = Z^{-1} \frac{\delta^2 Z}{\delta \bar{K}(x)\delta K(y)} \Big|_{K=0} = G_{xy} + (hG)_x (G\bar{h})_y. \tag{A.5}$$

To take the average we simply Wick contract $h$ and $\bar{h}$ with

$$\overline{h(x)\bar{h}(y)} = v\delta^{(d)}(x-y). \tag{A.6}$$

Then

$$\overline{\langle \phi(x) \rangle} = \overline{\langle \phi(x)\phi(y) \rangle} = 0, \tag{A.7}$$

consistently with the $U(1)$ symmetry being recovered on average. The non vanishing two-point function is

$$\overline{\langle \bar{\phi}(x)\phi(y) \rangle} = G_{xy} + v(GG)_{xy}. \tag{A.8}$$

The explicitly broken Ward identities for a $U(1)$ transformation read

$$\langle \partial_\mu J^\mu(x)\phi(y)\bar{\phi}(z) \rangle = \delta^{(d)}(x-y)\langle \phi(y)\bar{\phi}(z) \rangle - \delta^{(d)}(x-z)\langle \phi(y)\bar{\phi}(z) \rangle$$
$$- h(x)\langle \phi(x)\phi(y)\bar{\phi}(z) \rangle + \bar{h}(x)\langle \bar{\phi}(x)\phi(y)\bar{\phi}(z) \rangle. \tag{A.9}$$

The last two correlators equal

$$\langle \phi(x)\phi(y)\bar{\phi}(z) \rangle = G_{xz}(G\bar{h})_y + G_{yz}(G\bar{h})_x + (hG)_z(G\bar{h})_y(G\bar{h})_x,$$
$$\langle \bar{\phi}(x)\phi(y)\bar{\phi}(z) \rangle = G_{xy}(hG)_z + G_{yz}(hG)_x + (hG)_z(G\bar{h})_y(hG)_x, \tag{A.10}$$

so that

$$\overline{h(x)\langle \phi(x)\phi(y)\bar{\phi}(z) \rangle} = v G_{xy} G_{xz} + v G_{yz} G(0) + v^2 (GG)_{yz} G(0) + v^2 (GG)_{xz} G_{xy},$$
$$\overline{\bar{h}(x)\langle \bar{\phi}(x)\phi(y)\bar{\phi}(z) \rangle} = v G_{xy} G_{xz} + v G_{yz} G(0) + v^2 (GG)_{yz} G(0) + v^2 (GG)_{xy} G_{xz}. \tag{A.11}$$

The average of (A.9) reads then

$$\overline{\langle \partial_\mu J^\mu(x)\phi(y)\bar{\phi}(z) \rangle} = \delta^{(d)}(x-y)\overline{\langle \phi(y)\bar{\phi}(z) \rangle} - \delta^{(d)}(x-z)\overline{\langle \phi(y)\bar{\phi}(z) \rangle}$$
$$- v^2 \left( (GG)_{xz} G_{xy} - (GG)_{xy} G_{xz} \right). \tag{A.12}$$

It is straightforward to check (A.12) by using the explicit form of $J_\mu = \bar{\phi}\partial_\mu\phi - \phi\partial_\mu\bar{\phi}$ and performing the Wick contractions. We can now explicitly check the disordered Ward identity (41). Using the equations of motion we have $\partial_\mu J^\mu = \left(\bar{h}(x)\bar{\phi}(x) - h(x)\phi(x)\right)$, so that

$$\langle\partial^\mu J_\mu(x)\rangle = \int d^d z \left(h(z)\bar{h}(x) - h(x)\bar{h}(z)\right) G_{xz}. \tag{A.13}$$

Equivalently we can directly compute

$$\langle J_\mu(x)\rangle = \int d^d w d^d z\, h(z)\bar{h}(w) \left(\partial_\mu^{(x)} G_{xz} G_{xw} - G_{xz}\partial_\mu^{(x)} G_{xw}\right), \tag{A.14}$$

and take a derivative. As expected from the recovery of translation invariance after the average we find $\overline{\langle\partial^\mu J_\mu\rangle} = 0$. However, due to the presence of $h$, inserting $\langle\partial^\mu J_\mu\rangle$ under the average modifies the correlators, in particular

$$\begin{aligned}
\overline{\langle\partial^\mu J_\mu(x)\rangle\langle\phi(y)\bar{\phi}(z)\rangle} &= \int d^d w\, G_{xw}\overline{\left(h(w)\bar{h}(x) - h(x)\bar{h}(w)\right)\langle\phi(y)\bar{\phi}(z)\rangle} \\
&= -v^2 \left((GG)_{xz} G_{xy} - (GG)_{xy} G_{xz}\right).
\end{aligned} \tag{A.15}$$

This precisely corresponds to the last term in the right hand side of (A.12). Therefore, using the improved current $\widetilde{J}_\mu := J_\mu - \langle J_\mu\rangle$, the Ward identity (A.12) becomes

$$\overline{\langle\partial_\mu\widetilde{J}^\mu(x;h(x))\phi(y)\bar{\phi}(z)\rangle} = \delta^{(d)}(x-y)\overline{\langle\phi(y)\bar{\phi}(z)\rangle} - \delta^{(d)}(x-z)\overline{\langle\phi(y)\bar{\phi}(z)\rangle}, \tag{A.16}$$

in agreement with (41) with $k = 2$ operators. From here one can reproduce the exponentiation procedure and determine the presence of a topological operator in the disordered theory.

### A.1.2 Ensemble average

When $h$ is a constant every member of the ensemble is translation invariant. Indeed the one point function of the scalar field is now a constant:

$$\langle\phi(x)\rangle = \bar{h}\int d^d y\, G_{xy} = \frac{\bar{h}}{m^2}. \tag{A.17}$$

Note that the mass acts as a IR regulator. The two point functions are

$$\begin{aligned}
\langle\phi(x)\phi(y)\rangle &= \bar{h}^2 \int d^d z d^d w\, G_{xz} G_{yw} = \frac{\bar{h}^2}{m^4}, \\
\langle\bar{\phi}(x)\phi(y)\rangle &= G_{xy} + |h|^2 \int d^d z d^d w\, G_{xz} G_{yw} = G_{xy} + \frac{|h|^2}{m^4}.
\end{aligned} \tag{A.18}$$

In agreement with the $U(1)$ average symmetry, the only non-vanishing average two point function is

$$\overline{\langle\bar{\phi}(x)\phi(y)\rangle} = G_{xy} + \frac{v}{m^4}. \tag{A.19}$$

The explicitly broken Ward identities are

$$\begin{aligned}
\langle\partial_\mu J^\mu(x)\phi(y)\bar{\phi}(z)\rangle = &\delta^{(d)}(x-y)\langle\phi(y)\bar{\phi}(z)\rangle - \delta^{(d)}(x-z)\langle\phi(y)\bar{\phi}(z)\rangle \\
&- h\langle\phi(x)\phi(y)\bar{\phi}(z)\rangle + \bar{h}\langle\bar{\phi}(x)\phi(y)\bar{\phi}(z)\rangle.
\end{aligned} \tag{A.20}$$

The operator

$$\partial^\mu J_\mu(x) + h\phi(x) - \bar{h}\bar{\phi}(x), \tag{A.21}$$

generates the Ward identities, and we can now explicitly check that it integrates to zero on the whole space. The left hand side of (A.20) vanishes when integrating $x$ over the whole space. For the last two terms in the right hand side we get

$$\langle \phi(x)\phi(y)\bar{\phi}(z) \rangle = \frac{\bar{h}}{m^2}\left(G_{xz} + G_{yz}\right) + \frac{h\bar{h}^2}{m^6},$$

$$\langle \bar{\phi}(x)\phi(y)\bar{\phi}(z) \rangle = \frac{h}{m^2}\left(G_{xy} + G_{yz}\right) + \frac{\bar{h}h^2}{m^6},$$

(A.22)

so that

$$\overline{h\langle \phi(x)\phi(y)\bar{\phi}(z) \rangle} - \overline{\bar{h}\langle \bar{\phi}(x)\phi(y)\bar{\phi}(z) \rangle} = \frac{v}{m^2}\left(G_{xz} - G_{xy}\right). \tag{A.23}$$

Then, by translation invariance, we have

$$\int d^d x \left( \overline{h\langle \phi(x)\phi(y)\bar{\phi}(z) \rangle} - \overline{\bar{h}\langle \bar{\phi}(x)\phi(y)\bar{\phi}(z) \rangle} \right) = \frac{v}{m^2} \int d^d x \left( G_{xz} - G_{xy} \right) = 0, \tag{A.24}$$

where the support of the integral needs to be the entire space. In this simple example we have chosen a scalar deformation so that Poincaré invariance remains always unbroken, no tensor operator can get a vev, and all complications arising from non-vanishing vevs disappear. For example, specifying (A.13) to the case of constant $h$ immediately gives $\langle \partial_\mu J^\mu \rangle = 0$.

We can also compute $\langle \bar{\phi}(x_1)\phi(x_2) \rangle$ when $X$ is a disconnected space. For example, if $X^{(d)} = X_1^{(d)} \sqcup X_2^{(d)}$, $x_1 \in X_1^{(d)}$ an $x_2 \in X_2^{(d)}$, (A.2) reads

$$Z[K_{1,2}, \bar{K}_{1,2}, h] = \exp\left( \sum_{i=1,2} \int_{X_i^{(d)}} d^d x_i d^d y_i (\bar{h} + \bar{K}_i(x_i)) G(x_i - y_i)(h + K_i(y_i)) \right), \tag{A.25}$$

and we get

$$\langle \bar{\phi}(x_1)\phi(x_2) \rangle_X = Z^{-1} \left. \frac{\delta^2 Z}{\delta \bar{K}_1(x_1)\delta K_2(x_2)} \right|_{K=0} = \langle \bar{\phi}(x_1) \rangle_{X_1} \langle \phi(x_2) \rangle_{X_2} = \frac{|h|^2}{m^4}, \tag{A.26}$$

namely only the disconnected part of the correlator contributes. Averaging on $h$ we have

$$\overline{\langle \bar{\phi}(x_1)\phi(x_2) \rangle_X} = \overline{\langle \bar{\phi}(x_1) \rangle_{X_1} \langle \phi(x_2) \rangle_{X_2}} = \frac{v}{m^4}. \tag{A.27}$$

We explicitly see that in both $X_1$ and $X_2$ the $U(1)$ symmetry is explicitly broken and conserved only globally over the entire space $X$.

## A.2 't Hooft anomalies from replicas

We check the matching of t'Hooft anomalies between the pure and disordered theory in the simple example of the $U(1)$ chiral anomaly in $4d$. As well-known, a free massless Weyl fermion $\psi$ in $4d$ suffers from a cubic 't Hooft anomaly, which in momentum space reads

$$p_1^\mu \langle J_\mu(p_1) J_\nu(p_2) J_\rho(p_3) \rangle = i \frac{k}{16\pi^3} \epsilon_{\nu\rho\alpha\beta} p_2^\alpha p_3^\beta, \tag{A.28}$$

where $k = 1$. We deform the theory with a space dependent complex mass term $m(x)$, which explicitly breaks the $U(1)$ symmetry down to fermion parity. However, if we sample $m(x)$ from a Gaussian distribution proportional to $\bar{m}(x)m(x)$, then the disordered theory recovers the $U(1)$ symmetry via the conserved current $\widetilde{J}_\mu$. Since $\langle \widetilde{J}_\mu \rangle = 0$ before averaging, we have

$$\overline{\langle \widetilde{J}_\mu(p_1)\widetilde{J}_\nu(p_2)\widetilde{J}_\rho(p_3) \rangle} = \overline{\langle \widetilde{J}_\mu(p_1)\widetilde{J}_\nu(p_2)\widetilde{J}_\rho(p_3) \rangle_c} = \overline{\langle J_\mu(p_1) J_\nu(p_2) J_\rho(p_3) \rangle_c}. \tag{A.29}$$

The last three-point function is most easily evaluated using the replica trick. The replicated theory has $n$ Weyl fermions with a quartic deformation (spinor indices omitted)

$$S_{\text{rep}} = \sum_{a=1}^{n} S_{0,a} + v^2 \sum_{a,b} \psi_a \psi_a \overline{\psi}_b \overline{\psi}_b, \tag{A.30}$$

which is invariant under the diagonal $U(1)_D$ symmetry, with conserved current

$$J_D^{\mu} = \sum_a J_a^{\mu}. \tag{A.31}$$

According to (79), we have

$$\overline{\langle J_{\mu}(p_1) J_{\nu}(p_2) J_{\rho}(p_3) \rangle_c} = \lim_{n \to 0} \frac{\partial}{\partial n} \langle J_{D,\mu}(p_1) J_{D,\nu}(p_2) J_{D,\rho}(p_3) \rangle^{\text{rep}}. \tag{A.32}$$

The $U(1)_D$ in the replica theory also suffers from a a cubic 't Hooft anomaly

$$p_1^{\mu} \langle J_{D,\mu}(p_1) J_{D,\nu}(p_2) J_{D,\rho}(p_3) \rangle_{\text{rep}} = \frac{ik}{16\pi^3} \epsilon_{\nu\rho\alpha\beta} p_2^{\alpha} p_3^{\beta}, \tag{A.33}$$

where $k = n$, since all $n$ fermions rotate (with the same charge) under $U(1)_D$. We then get

$$p_1^{\mu} \overline{\langle \widetilde{J}_{\mu}(p_1) \widetilde{J}_{\nu}(p_2) \widetilde{J}_{\rho}(p_3) \rangle} = \lim_{n \to 0} \frac{\partial}{\partial n} \left( \frac{in}{16\pi^3} \epsilon_{\nu\rho\alpha\beta} p_2^{\alpha} p_3^{\beta} \right) = \frac{i}{16\pi^3} \epsilon_{\nu\rho\alpha\beta} p_2^{\alpha} p_3^{\beta}, \tag{A.34}$$

which shows that the anomaly of the pure theory persists after the quenched average and also affects the disordered symmetry, in agreement with the results in the main text.

# B  Symmetry operators for averaged symmetries

In this appendix we prove the existence, and explicitly construct, an operator $\widehat{U}_g$ which implements the action of the group rather than the action of the corresponding Lie algebra for average symmetries. To this purpose we need to find an infinite set of operators $\widehat{Q}_n$ which have the same properties of $\widehat{Q}$ defined in (134) and which satisfy the identities

$$\langle \widehat{Q}_n \mathcal{O}_1 \cdots \mathcal{O}_k \rangle = \chi^n \left( \Sigma^{(d-1)} \right) \langle \mathcal{O}_1 \cdots \mathcal{O}_k \rangle, \qquad \forall\, n \in \mathbb{N}, \tag{B.1}$$

where we recall that $\chi(\Sigma^{(d-1)})$ denotes the sum of the charges of the local operators which are inside the surface $\Sigma^{(d-1)}$. Note that (B.1) applies before ensemble averaging. We define $\widehat{Q}_0 = 1$ and $\widehat{Q}_1 = \widehat{Q}$. We find $\widehat{Q}_n[\Sigma^{(d-1)}, D^{(d)}; h]$ for $n > 1$ iteratively. Suppose that there exists an operator $\widehat{Q}_{n-1}$ such that

$$\langle \widehat{Q}_{n-1} \Phi \rangle = \chi^{n-1}(\Sigma^{(d-1)}) \langle \Phi \rangle, \tag{B.2}$$

for any product of local operators $\Phi$. We then compute

$$\langle \widehat{Q}_{n-1} \widehat{Q}_1 \Phi \rangle = \chi^{n-1} \langle Q\Phi \rangle + q_0 (\chi + q_0)^{n-1} \left\langle h \int_{D^{(d)}} \mathcal{O}_0(x) \Phi \right\rangle - q_0 (\chi - q_0)^{n-1} \left\langle \overline{h} \int_{D^{(d)}} \overline{\mathcal{O}}_0(x) \Phi \right\rangle$$

$$= \chi^n \overline{\langle \Phi \rangle} + q_0 \sum_{k=0}^{n-2} \binom{n-1}{k} \chi^k q_0^{n-1-k} \left( \left\langle h \int_{D^{(d)}} \mathcal{O}_0(x) \Phi \right\rangle - (-1)^{n-k-1} \left\langle \overline{h} \int_{D^{(d)}} \overline{\mathcal{O}}_0(x) \Phi \right\rangle \right). \tag{B.3}$$

Next we introduce operators $\Gamma_l$ defined in such a way that

$$\left\langle \Gamma_l \int_{D^{(d)}} \mathcal{O}_0(x) \Phi \right\rangle = \chi^l \left\langle \int_{D^{(d)}} \mathcal{O}_0(x) \Phi \right\rangle. \tag{B.4}$$

Their existence follows from the (by now assumed) existence of the operators $\widehat{Q}_n$. In fact, it is easy to see that the $\Gamma_l$'s satisfy the relation

$$\widehat{Q}_l = \sum_{s=0}^{l}\binom{l}{s}q_0^{l-s}\Gamma_s\,,\tag{B.5}$$

valid when inserted in (vacuum to vacuum) correlators of the form $\langle \int_{D^{(d)}} \mathcal{O}_0(x)\Phi\rangle$. Now consider the vectors $\widehat{\mathbf{Q}} = (\widehat{Q}_0, \widehat{Q}_1, \cdots, \widehat{Q}_N)$ and $\mathbf{\Gamma} = (\Gamma_0, \Gamma_1, \cdots, \Gamma_N)$, with $\Gamma_0 = 1$. These are related as $\widehat{\mathbf{Q}} = A \cdot \mathbf{\Gamma}$ where $A = \mathbb{1} + T$ and $T$ is a strictly lower triangular matrix with non-vanishing entries

$$T_{l,s} = \binom{l}{s}q_0^{l-s}\,.\tag{B.6}$$

We can invert (B.5) as

$$\Gamma_l = \sum_{s=0}^{l} A_{l,s}^{-1}\widehat{Q}_s\,,\tag{B.7}$$

where we used that

$$A^{-1} = \mathbb{1} + \sum_{i=1}^{N}(-1)^i T^i\,,\tag{B.8}$$

is again a lower triangular matrix. An analogous analysis can be carried out for the operators $\overline{\Gamma}_l$ defined by

$$\left\langle \overline{\Gamma}_l \overline{h} \int_{D^{(d)}} \overline{\mathcal{O}_0}(x)\Phi \right\rangle = \chi^l \left\langle \overline{h} \int_{D^{(d)}} \overline{\mathcal{O}}_0(x)\Phi \right\rangle\,,\tag{B.9}$$

by simply replacing $q_0$ with $-q_0$, and we define $\overline{A}$ as $\widehat{\mathbf{Q}} = \overline{A} \cdot \overline{\mathbf{\Gamma}}$. We rewrite (B.3) as

$$\begin{aligned}
\langle \widehat{Q}_{n-1}\widehat{Q}_1\Phi\rangle &= \langle \widehat{Q}_n\Phi\rangle \\
&+ q_0 \sum_{k=0}^{n-2}\binom{n-1}{k}q_0^{n-1-k}\left(\left\langle \Gamma_k h \int_{D^{(d)}} \mathcal{O}_0(x)\Phi\right\rangle - (-1)^{n-1-k}\left\langle \overline{\Gamma}_k\overline{h}\int_{D^{(d)}}\overline{\mathcal{O}_0}(x)\Phi\right\rangle\right) \\
&= \langle\widehat{Q}_n\Phi\rangle + q_0\left[\left\langle \widehat{Q}_{n-1}\left(h\int_{D^{(d)}}\mathcal{O}_0(x) - \overline{h}\int_{D^{(d)}}\overline{\mathcal{O}_0}(x)\right)\Phi\right\rangle\right. \\
&\left. + \left\langle \Gamma_{n-1}h\int_{D^{(d)}}\mathcal{O}_0(x)\Phi\right\rangle - \left\langle\overline{\Gamma}_{n-1}\overline{h}\int_{D^{(d)}}\overline{\mathcal{O}_0}(x)\Phi\right\rangle\right] \\
&= \langle\widehat{Q}_n\Phi\rangle + q_0\left[\left\langle\widehat{Q}_{n-1}\left(h\int_{D^{(d)}}\mathcal{O}_0(x) - \overline{h}\int_{D^{(d)}}\overline{\mathcal{O}_0}(x)\right)\Phi\right\rangle\right. \\
&\left. + \sum_{k=0}^{n-1}\widehat{Q}_k\left(\left\langle A_{n-1,k}^{-1}h\int_{D^{(d)}}\mathcal{O}_0(x)\Phi\right\rangle - \left\langle\overline{A}_{n-1,k}^{-1}\overline{h}\int_{D^{(d)}}\overline{\mathcal{O}_0}(x)\Phi\right\rangle\right)\right] \\
&= \langle\widehat{Q}_n\Phi\rangle + q_0\sum_{k=0}^{n-2}\widehat{Q}_k\left[\left\langle A_{n-1,k}^{-1}h\int_{D^{(d)}}\mathcal{O}_0(x)\Phi\right\rangle - \left\langle\overline{A}_{n-1,k}^{-1}\overline{h}\int_{D^{(d)}}\overline{\mathcal{O}_0}(x)\Phi\right\rangle\right]\,,
\end{aligned}\tag{B.10}$$

where we used that $A_{k,k}^{-1} = \overline{A}_{k,k}^{-1} = 1$. We then find the recursion relation

$$\widehat{Q}_n = \widehat{Q}_{n-1}\widehat{Q}_1 - q_0\sum_{k=0}^{n-2}\widehat{Q}_k\left(A_{n-1,k}^{-1}h\int_{D^{(d)}}\mathcal{O}_0(x) - \overline{A}_{n-1,k}^{-1}\overline{h}\int_{D^{(d)}}\overline{\mathcal{O}_0}\right)\,,\tag{B.11}$$

which proves the existence of $\widehat{Q}_n$ for every values of $n \in \mathbb{N}$.

As an example consider $N = 3$. We have

$$
A = \begin{pmatrix} 1 & 0 & 0 & 0 \\ q_0 & 1 & 0 & 0 \\ q_0^2 & 2q_0 & 1 & 0 \\ q_0^3 & 3q_0^2 & 3q_0 & 1 \end{pmatrix}, \qquad A^{-1} = \begin{pmatrix} 1 & 0 & 0 & 0 \\ -q_0 & 1 & 0 & 0 \\ q_0^2 & -2q_0 & 1 & 0 \\ -q_0^3 & 3q_0^2 & -3q_0 & 1 \end{pmatrix}, \tag{B.12}
$$

and

$$
\widehat{Q}_2 = \widehat{Q}_1^2 + q_0^2 \left( h \int_{D^{(d)}} \mathcal{O}_0(x) + \bar{h} \int_{D^{(d)}} \overline{\mathcal{O}_0} \right),
$$

$$
\widehat{Q}_3 = \widehat{Q}_2 \widehat{Q}_1 - q_0^3 \int_{D^{(d)}} \mathcal{D}(x) - 2q_0^2 \widehat{Q}_1 \left( h \int_{D^{(d)}} \mathcal{O}_0 + \bar{h} \int_{D^{(d)}} \overline{\mathcal{O}_0} \right). \tag{B.13}
$$

We now crucially verify that the charges $\widehat{Q}_n$ vanish when $D^{(d)} = X^{(d)}$ *after* ensemble average in arbitrary local correlators. For this purpose we derive a further constraint on correlators involving arbitrary functions of $h$ and $\bar{h}$. Consider

$$
\int dh \, d\bar{h} \, P[\bar{h}h] f(h, \bar{h}) \frac{\int \mathcal{D}\mu \, e^{-S_0 - (h \int \mathcal{O}_0 + c.c.) + \int K_i \mathcal{O}_i}}{\int \mathcal{D}\mu \, e^{-S_0 - (h \int \mathcal{O}_0 + c.c.)}}, \tag{B.14}
$$

where $f$ is an arbitrary smooth function of $h$ and $\bar{h}$. We shift $h \to h + \epsilon \delta h$, where $\delta h = -iq_0 h$. Using that $\delta h \mathcal{O}_0 = -h \delta \mathcal{O}_0$ and expanding to linear order in $\epsilon$ we get[27]

$$
iq_0 f(h, \bar{h}) \overline{\langle \int_{X^{(d)}} \mathcal{D}(x; h) \mathcal{O}_1 \cdots \mathcal{O}_n \rangle} = -\overline{\delta f(h, \bar{h}) \langle \mathcal{O}_1 \cdots \mathcal{O}_n \rangle}, \tag{B.15}
$$

where

$$
\delta f(h, \bar{h}) = \partial f \, \delta h + \overline{\partial} f \, \delta \bar{h}. \tag{B.16}
$$

Thanks to (B.15) we can now show that

$$
\overline{\langle \widehat{Q}_n[\emptyset, X^{(d)}; h] \Phi \rangle} = 0, \qquad n > 0. \tag{B.17}
$$

Let us explicitly work out the $n = 2, 3$ cases. For $n = 2$ it is enough to use (B.15) with $f = h$ and $f = \bar{h}$ to get the identity

$$
\overline{\left\langle \left( \int_{X^{(d)}} \mathcal{D}(x; h) \right)^2 \Phi \right\rangle} + \overline{\left\langle \left( h \int_{X^{(d)}} \mathcal{O}_0(x) + \bar{h} \int_{X^{(d)}} \overline{\mathcal{O}_0} \right) \Phi \right\rangle} = 0. \tag{B.18}
$$

We can plug this relation into $\widehat{Q}_2$ in (B.13) to immediately get (B.17) for $n = 2$. For $n = 3$ we use (B.15) with the functions $h^2, \bar{h}^2$ and $h\bar{h}$. In this way we get the relations

$$
\overline{\left\langle \left( \int_{X^{(d)}} \mathcal{D}(x) \right)^3 \Phi \right\rangle} = 2 \overline{\left\langle \left( \int_{X^{(d)}} \mathcal{D}(x; h) \right) \left( h \int_{X^{(d)}} \mathcal{O}_0 + \bar{h} \int_{X^{(d)}} \overline{\mathcal{O}_0} \right) \Phi \right\rangle} = 0, \tag{B.19}
$$

which, pluggged in $\widehat{Q}_3$ in (B.13) allows us to get (B.17) for $n = 3$. We can then construct the non-genuine symmetry operator

$$
\widehat{U}_g \left[ \Sigma^{(d-1)}, D^{(d)}; h \right] = \sum_{n=0}^{\infty} \frac{(i\alpha)^n}{n!} \widehat{Q}_n \left[ \Sigma^{(d-1)}, D^{(d)}; h \right], \qquad g = e^{i\alpha}, \tag{B.20}
$$

which, similarly to $\widehat{Q}[\Sigma^{(d-1)}, D^{(d)}; h]$, becomes quasi-genuine when $D^{(d)} = X^{(d)}$.

We have then shown the existence, and explicitly constructed, the operator $\widehat{U}_g$ which implements the selection rules imposed by the emergent symmetries.

---

[27]An extra term coming from the denominator of (B.14) vanishes because of (131).

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
