# Peer review of "Symmetries and topological operators, on average"

_SciPost Physics, doi:SciPost Phys. 15, 125 (2023)_

## Round 1 · Referee Report · Anonymous (Referee 1) · 2023-6-21

Report

The paper studies what happens when a pure model is coupled to a random variable that explicitly breaks a symmetry which however re-emerges after averaging. The main goal of the paper is to define how the resulting symmetries define selection rules and Ward identities, and identify the associated topological operators. The authors study two separate cases that they dub “disordered symmetries” and “averaged symmetries”. The former type of symmetries appear when the random couplings posses an explicit space dependence. The second ones arise when studying random couplings which do not depend on the position. They show how the Ward identities are modified in both cases. For disordered symmetries they also explain how to recover the same result from the replicated formulation of the disordered model. While for disordered symmetries they find a topological operator associated to a given symmetry, they show that for averaged symmetries no (genuine) topological operator can be written. This implies that the latter symmetries cannot be gauged, which matches with recent AdS/CFT lore that the bulk dual of these symmetries are global symmetries (non-perturbatively broken by wormhole configurations) and not gauge symmetries.

The paper is clear and well written. The question that is studied is a very fundamental one in the study of disordered theories. I thus believe that the paper should be published. I suggest some minor revisions.

A few small remarks: 1) In page 3: “For these reasons, the idea of symmetry has recently been made more precise and intrinsic through the notion of topological operators [1]. ” The fact that [1] is cited for the definition of topological operators seems a bit excessive, meaning that topological operators are textbook material at least for the case of usual (0-form) symmetries which is the focus of the paper. Maybe a softer citation would be more appropriate? E.g. “The idea of symmetry can be made more precise and intrinsic through the notion of topological operators (see [1] for a modern treatment).” 2) The second term in (2.11) can be divergent (e.g. if $S_0$ is a CFT) and typically one assumes a regularisation scheme to keep the result finite. If this was assumed it would not hurt to mention. 3) Below (3.12): “are always expressed“ $\to$ “can always be expressed” 4) Above (2.35): “By changing variables in the path integral”. It would be useful to refer to formula (2.10) otherwise it may seem like a new computation. Maybe it could be rephrased as “From (2.10) with no $\mathcal O_1 \dots \mathcal O_k$ insertions”? 5) Last comment of page 15. Only translations are mentioned but it may be worth mentioning that the same holds for rotations. 6) Sometimes the notation is rather obvious but not explicitly defined. The authors may think about adding short definitions in such cases, e.g. a) $R^\vee_i$ in (2.25) b) Subscript $K$ of $\langle \cdot \rangle_K$ in (2.30) c) $X^{(d)}$ and $\alpha(\lambda,A)$ in (2.61) d) $q_j^{(m)},K_j^{(m)}$ in (4.14) e) $M^{d+1}$ and $X_i^{(d)}$ in Figure 3 f) maybe even the "obvious notation" of (5.3)?

A couple of typos: - Below (2.50): "an other" $\to$ "another" - (4.6): "$ q_{i,S} - q_{i,F}$" $\to$ "$ q_{S,i} - q_{F,i}$"

Question/curiosity: Page 15 second bullet. The authors say that $\tilde J$ has a $h$ dependent anomalous dimension that averages to zero. Did you check how this cancellation happens in an example (in perturbation theory), is it a straightforward mechanism or it requires a non-trivial conspiracy of terms?

  • validity: -
  • significance: -
  • originality: -
  • clarity: -
  • formatting: -
  • grammar: -

Author:  Marco Serone  on 2023-08-03  [id 3872]

(in reply to Report 2 on 2023-06-21)

We thank the referee for the positive report and the useful suggestions.

We agree on all the remarks and suggestions made, which have been implemented in the revised version.
In particular, point 2) has been addressed by adding a footnote shortly after (2.11) and upgrading an old footnote in the main text to avoid
too many footnotes.

Regarding the question/curiosity: from the example of a random field free scalar discussed in App.A, we believe that
cancellation (on average) of the anomalous dimension of the improved current would arise from symmetry reasons (after average over $h$)
rather than non-trivial conspiracies of terms.
But this is only a speculation, because we did not check explicitly the cancellation in any given example.

---

## Round 1 · Referee Report · Slava Rychkov (Referee 3) · 2023-7-17

Report

The paper studies how, in theories with quenched disorder, symmetries broken for any specific realization of disorder, get restored after averaging over disorder distribution. This is an interesting and timely topic. The paper is written following the modern point of view associating symmetries with topological defect line operators. This appears to be the first paper discussing symmetries in disordered systems in this language.

I recommend the paper to publication. Below is a list of remarks which the authors might wish to consider for the revised version.

General remark: I found it confusing that the dependence of various quantities on h is not made explicit. E.g. the current is h independent, but after it is corrected it becomes h-dependent, while most other operators one talks about are h-independent.

  • it's not clear to me how the conclusion below 2.11 follows from the given equations. Is there a mathematical argument which shows that things indeed happen the way the authors claim they happen? (Although I agree that on general grounds things SHOULD happen this way.) The second term in the r.h.s. of 2.11 varies monotonically with the increase of the surface, why would it tend to zero as the surface is getting larger?

  • section 2.2, second line - are references 64,65 really the original references for this basic idea?

  • p.11 - "when 2.15 is satisfied..." This sentence is true only for small disorder. When disorder is large, new disordered fixed points may appear even if small disorder is irrelevant. A classic example is the Nishimori point, see e.g. "NISHIMORI POINT IN RANDOM-BOND ISING AND POTTS MODELS IN 2D" by Honecker et al.

  • p.13 last paragraph - here a reference to mean field theory examples which satisfy selection rules but have no currents nor topological defect lines could be appropriate.

  • p.14 "an denominator" -> "and denominator", "setting them" -> "setting $K_i$" (otherwise ambiguous)

  • p.14 construction of the modified current raises a question: why such a current would satisfy OPE of the same form as currents usually satisfy?

  • p.16 "an higher-group" -> "a higher-group"

  • p.19 I guess a reference to appendix B might be needed

  • p.24 refs to cond.mat. phases as an example for the theory developed in this paper appear inappropriate, since the disorder in cond.mat.systems is time-independent. Indeed the authors are aware of this, in footnote 11, but then why do they cite 47-51,53,52? at least an explanation is warranted.

  • p.30 "exotic selection rules" why exotic? to me they appear completely natural, as they would to anyone who spent any time thinking about theories with random field disorder. anyway they are quite standard in this context.

G semidirect $S_n$ -> should it be $G^n$ semidirect $S_n$ (also on p.31)

  • p.31 "assuming that $J^\mu_D$ is conserved in the IR also at finite $n$"

It is a big assumption that the RG flow reaches a fixed point, so that the symmetry has a chance to emerge, also at finite $n$. It would be great to warn the reader about one example when this does not happen - the Random Field Ising Model, as is discussed e.g. in my papers with Trevisani and Kaviraj. The fixed point in that model exists only at $n=0$ (2009.10087, Section 6). The $n=0$ fixed point is not a logarithmic CFT, but an ordinary CFT (1912.01617), enriched with Parisi-Sourlas supersymmetry. This is what happens in the free theory and close to the upper critical dimension of the interacting theory.

  • p.33 - the previous remark applies.

The first sentence of Section 4.2 is ambiguous. "Can" or "may"? What was discussed in [56,57], logCFTs themselves or that IR fixed points of theories with disorder can be described by them?

-p 44 - the discussion in the conclusions would benefit from comparison with Random Field Ising Model. In this theory, as recently emphasized in my work with Kaviraj and Trevisani, operators with well-defined scaling dimension do not form multiplets of $S_n$, in the limit $n\to 0$, yet the $S_n$ invariance is not spontaneously broken.

  • validity: -
  • significance: -
  • originality: -
  • clarity: -
  • formatting: -
  • grammar: -

Author:  Marco Serone  on 2023-08-03  [id 3870]

(in reply to Report 1 by Slava Rychkov on 2023-07-17)

We thank the referee for the positive report and the useful suggestions.

Below our detailed reply, where each hyphen refers to the one in the referee's report in sequential order.

General remark. We agree on making explicit the $h$ dependence of the various tilded/hatted quantities, and we implemented it in the revised version.

  • The second term in the r.h.s. generally does not go to zero as the surface increases, and we are not claiming that. When the deformation is irrelevant (and not just classically irrelevant) that term is expected to be suppressed with respect to the first one in the r.h.s. at large distances, i.e. when the $k$ external operators are far apart. We did not mean to prove the above statement below (2.11), so we added "expected" in the text to make it clear that this is simply the usual lore of what emergent symmetry means in the context of Ward identities.

  • These are not the original references and we accordingly fixed the citations.

  • We agree with the referee and changed accordingly the text below (2.15) and added a footnote referring to the work of Nishimori.

  • We agree that generalized free or mean field theories do not have local conserved currents which, at least in a standard sense, lead to a topological operator. On the other hand these theories are non-local and may have non-local conserved currents. For this reason we prefer not to mention them here and keep the discussion within the framework of local quantum field theories for simplicity.

  • corrected.

  • We believe that the modified current satisfies a different OPE with respect to ordinary currents but reduces to the same after quenched average.
    We have however not investigated the details since OPE is not crucial for the considerations of the paper.

  • corrected.

  • App. B only deals with the case of constant $h$. In p.19, before (2.52), we refer to unpublished checks that we carried out.

  • The referee is right that the disorder in our paper and that in refs.[48-55] is different. We cited those works only in connection with the existence of a topological protection mechanism, 't Hooft anomaly in the euclidean theory in our case, SPT in the refs. cited, in presence of disorder. 't Hooft anomalies and SPT phases in pure theories are not independent concepts and we believe that our continuum description, applied to time-independent disorder, might lead to a different perspective to the works [48-55].

  • "Exotic" with respect to standard unitary QFTs and, we believe, for readers with less familiarity than the referee to QFTs with disorder.

In the replica theory the symmetry is actually $G\rtimes S_n$ and not $G^n \rtimes S_n$, where $G$ is the diagonal subgroup of $G^n$, because the remaining $G^{n-1}$ is broken by disorder induced interactions.

  • We added a footnote before (4.1) pointing out the non-triviality of the assumption and a ref. to the works cited by the referee.

We rephrased the first sentence of Section 4.2 to avoid the ambiguity pointed out by the referee.

  • We believe that referring in the conclusions to the replica theory of the Random Field Ising model, which does not have a smooth $n\rightarrow 0$ limit, might be confusing, as we assume analyticity at $n=0$ in the discussion there. The peculiarity of this model has been however pointed out in the new footnote 17.

---

## Round 1 · Referee Report · Anonymous (Referee 2) · 2023-7-17

Strengths

The paper extensively studies aspects of symmetries that are broken in individual members of the ensemble but arise in disordered theories after averaging. Such an analysis is performed both in theories that have a space-dependent random coupling and a constant random coupling. Such symmetries have received a lot of attention both in condensed matter physics (see 2305.16399 for a more recent review) and in quantum gravity (see 2011.09444, which conjectured that such emergent symmetries are dual to bulk global symmetries instead of bulk gauge symmetries in a putative gravitational dual). Phrasing such symmetries in the modern language of topological operators and analyzing the Ward identities of such symmetries, as done in the paper under review, is a tremendous contribution to these fields.

Weaknesses

While phrasing such symmetries in a formal language is important, since I believe that this paper has the potential to become a "cornerstone" reference for all future researchers that will study symmetries in disordered theories, I believe that the paper should be slightly more pedagogical. Specifically, I believe that it lacks specific examples for the numerous concepts introduced throughout the paper. For instance, in section 2, can the authors provide an example where `t Hooft anomalies can be harnessed to constrain the RG flow of a theory with quenched disorder? Working out such an example will not only be pedagogically useful but will help emphasize that the new tools developed by the authors are useful in practice. Similarly, in section 5, the specific example of a theory with a constant random coupling would be useful to discuss in parallel to the discussion of Ward identities and topological operators. For instance, the authors can consider the example of the SYK model (for the purpose of Ward identities) or the example of ensemble average d 2d CFTs to exemplify why the topological operator cannot be placed on a homologically non-trivial surface. Regarding the gravity discussion presented in section 5.3, I believe the authors should discuss further why their results support (or not) the conjecture that emergent symmetries after ensemble averaging are dual to bulk global symmetries. The authors note that their results are consistent with the violation of charge conservation due to wormhole geometries (already discussed in references 60-62) but do not study the implications of their analysis in further detail. I believe that addressing the following questions would help strengthen the paper. What is the bulk dual of the boundary topological operators that the authors discuss? Is the fact that the boundary topological operators cannot be placed on homologically non-trivial cycles consistent with this conjecture? Presumably, the answer to this latter question is related to the fact that something goes wrong with the bulk dual of such topological operators when summing over all bulk geometries, including wormhole geometries.

Finally, I believe the authors should also further discuss which ones of their results also apply to higher-form and non-invertible symmetries. The authors make a few comments about higher-form symmetries throughout the paper, but summarizing which results are applicable in the discussion would be useful to the reader. It would also be useful to give examples where such symmetries actually arise after taking an ensemble average: as far as I know, no such examples are presented in the literature (both for higher-form and for non-invertible symmetries).

Report

As discussed above, I believe the paper will serve as a cornerstone reference for future work on emergent symmetries in disordered theories. Therefore, I strongly recommend that this paper be published in SciPost without hesitation. I believe the main modifications that will greatly strengthen the paper are in providing the examples that I requested above and in further discussing the implications of the findings from section 5.1 and 5.2 in a putative gravitational dual theory.

Requested changes

  1. Provide an example of `t Hooft anomaly matching in section 2.

  2. Study the Ward identity and topological operators in section 5.1 and section 5.2 in parallel to the study of a specific example where a symmetry is emergent after an ensemble average.

  3. Answer the question posed above about the meaning of the boundary topological operators in a putative bulk dual.

  4. Can the authors expand on why their results are compatible with the results of [82]. I did not fully understand the last paragraph in the conclusion.

  5. Discuss how the results in this paper are applicable to higher-form and non-invertible symmetries. This would also help further the discussion about the completeness of the charge spectrum in quantum gravity that is discussed at the end of section 5.3.

  • validity: top
  • significance: high
  • originality: high
  • clarity: high
  • formatting: excellent
  • grammar: excellent

Author:  Marco Serone  on 2023-08-03  [id 3871]

(in reply to Report 3 on 2023-07-17)

We thank the referee for the positive report and the useful suggestions.

  1. We actually did have provided an example (though a simple one) of 't Hooft anomaly matching in section 2. This is briefly discussed in section 2.4 and further details are found in appendix A.2. After the referee request, we realized that this point was not sufficiently emphasized in the paper, so we expanded the discussion concerning anomaly matching at the end of section 2.4.

  2. We provide a very simple example of Ward identities in App.A. We choose such a trivial case for the sake of explicitness and to avoid complications that would obscure the results.

  3. This is a great question, in our work however we want to be general and not commit to specific examples. Indeed one might speculate about the bulk dual of topological operators but this could be highly model-dependent.
    Moreover our analysis is based on a model of the boundary, rather than bulk, theory. In this set-up we can certainly argue about some "universal" aspects of the bulk theory but probably a more detailed understanding of the bulk physics would require further input data. These questions are very interesting and we are planning to investigate them in future works.

  4. We suspect the referee was referring to ref.[83] (Benini et al.) rather than ref.[82] (Heidenreich et al.), numbering as in first version of the paper. The theories considered in Benini et al. fall outside of our set-up and in this sense their results are trivially compatible with ours. We clarified this point at the end of the conclusions.

  5. Generalizations to higher-form symmetries are non trivial, as they cannot be broken by adding local operators and hence they are never broken by disorder. On the other hand, for 0-form non-invertible symmetries broken by random interaction many of our considerations should apply. We thought about some examples but we did not include them in the paper since the only conclusions we found were rather trivial. For instance the Ising CFT perturbed with random energy operator (which is charged under the Kramers-Wannier duality line) has the non-invertible symmetry. However the perturbation is marginally irrelevant and the IR is again the Ising CFT. It is a very interesting problem for the future to find non-trivial examples. The above considerations are briefly discussed in the next to last paragraph of the conclusions.

---

## Editorial Decision

published